# Comprehensive promotion of iPSC-CM maturation by integrating metabolic medium with nanopatterning and electrostimulation

Wener Li[1], Xiaojing Luo[1], Anna Strano[1], Shakthi Arun[1], Oliver Gamm[1], Mareike S. Poetsch[1], Marcel Hasse[1], Robert-Patrick Steiner[1], Konstanze Fischer[1], Jessie Pöche[1], Ying Ulbricht[1], Mathias Lesche[2], Giulia Trimaglio[3,4], Ali El-Armouche[1], Andreas Dahl[2], Peter Mirtschink[3], Kaomei Guan[1,5]✉ & Mario Schubert[1,5]✉

The immaturity of human induced pluripotent stem cell-derived cardiomyocytes (iPSC-CMs) is a major limitation for their use in drug screening to identify pro-arrhythmogenic or cardiotoxic molecules. Here, we demonstrate an approach that combines lipid-enriched maturation medium with a high concentration of calcium, nanopatterning of culture surfaces and electrostimulation to generate iPSC-CMs with advanced electrophysiological, structural and metabolic phenotypes. Systematic testing reveals that electrostimulation is the key driver of enhanced mitochondrial development and metabolic maturation and improved electrophysiological properties of iPSC-CMs. Increased calcium concentration strongly promotes electrophysiological maturation, while nanopatterning primarily facilitates sarcomere organisation with minor effect on electrophysiological properties. Transcriptome analysis reveals that activation of HMCES and TFAM targets contributes to mitochondrial development, whereas downregulation of MAPK/PI3K and SRF targets is associated with iPSC-CM polyploidy. These findings provide mechanistic insights into iPSC-CM maturation, paving the way for pharmacological responses that more closely resemble those of adult CMs.

The discovery of human induced pluripotent stem cells (iPSCs) represents a breakthrough for medical research and clinical application. As an unlimited source of cardiomyocytes (CMs) with a patient-specific genetic background, iPSC-derived CMs (iPSC-CMs) can be employed to explore disease mechanisms and drug effects. Additionally, they hold the potential for regenerating lost myocardium in patients with heart failure[1]. Numerous studies have demonstrated the ability of iPSC-CMs to recapitulate clinical features of inherited cardiomyopathies[2], arrhythmias[3,4] and cardiotoxic drug responses[5], even with patient-specific sensitivities[6-8]. Despite these achievements,

[1]Institute of Pharmacology and Toxicology, Technische Universität Dresden, Dresden, Germany. [2]DRESDEN-concept Genome Center, Center for Molecular and Cellular Bioengineering, Technische Universität Dresden, Dresden, Germany. [3]Institute of Clinical Chemistry and Laboratory Medicine, Department of Clinical Pathobiochemistry, University Hospital Dresden, Dresden, Germany. [4]National Center for Tumor Diseases, Partner Site Dresden, 01307 Dresden, and German Cancer Research Center, Heidelberg, Germany. [5]These authors jointly supervised this work: Kaomei Guan, Mario Schubert. ✉e-mail: kaomei.guan@tu-dresden.de; mario.schubert1@tu-dresden.de

their immaturity remains a significant limitation. In comparison to adult CMs, iPSC-CMs exhibit considerable differences in their morphology, gene expression patterns, metabolism and functionality[1,9], which may explain their low sensitivity to hypoxia(-reperfusion) injury[10,11], or the lack of features expected from a clinical phenotype of an inherited disease[12,13]. The use of iPSC-CMs to predict the pro-arrhythmic activity of drugs in the Comprehensive in vitro Proarrhythmia Assay (CiPA) has shown good correlation with the clinical risk of torsade de pointes or QTc prolongation. However, discrepancies have been reported for some multichannel blockers. For example, although verapamil has a good safety profile in the clinic, it abolishes the beating activity of iPSC-CMs at clinically relevant concentrations[14], probably due to differences in the expression of genes encoding ion channels, such as *SCN5A* ($Na_V1.5$), *CACNA1C* ($Ca_V1.2$), *KCNH2* (hERG), and *KCNQ1* ($K_V7.1$)[15], and in calcium handling, as well as less structural maturity (e.g., fewer mitochondrial networks, lack of T-tubules and intercalated discs)[1,9] in iPSC-CMs compared with adult CMs. Whether the establishment of more adult-like current patterns in iPSC-CMs affects their drug response remains elusive[16].

In recent years, several approaches have been developed to improve the maturation of iPSC-CMs. The integration of fibroblasts and/or endothelial cells into engineered heart tissue (EHT) models using iPSC-CMs significantly enhanced both structural and functional development, as demonstrated in various studies[17,18]. However, it is worth noting that 3D-tissue generation is challenging and the experimental throughput is lower compared to 2D cultures[19]. Other approaches to enhance iPSC-CM maturation include supplementation of the culture medium with fatty acids (FA)[13,20–22], hormones or small molecules[10,23], micro- or nanopatterning (NP) of culture surfaces[24,25], and electrostimulation (ES)[26–30]. As these stimuli have been investigated independently, the most effective factor for enhancing iPSC-CM maturation remains unclear. It is uncertain whether combined approaches could produce synergistic effects, and the underlying mechanisms driving the advanced maturation of iPSC-CMs remain to be elucidated.

Here, we systematically examined the effects of FA-enriched maturation medium (MM), increased calcium concentrations in the medium, NP and ES on the electrophysiological, structural and metabolic maturation of iPSC-CMs through comprehensive functional and molecular analyses. While NP primarily promoted structural maturation with limited impact on electrophysiological properties, elevated $Ca^{2+}$ concentrations in the medium strongly influenced the electrophysiological characteristics of iPSC-CMs. Notably, ES emerged as the key driver of enhanced mitochondrial development and metabolic maturation and improved electrophysiological properties. The combined application of MM with a high calcium concentration, NP and ES led to significant changes in the sensitivity of iPSC-CMs to cardioactive drugs, yielding pharmacological responses that more closely resemble those of adult CMs. Our data provide mechanistic insights into the distinct roles of these stimuli in shaping the cellular structure, metabolism and electrophysiology of iPSC-CMs.

## Results
We used the directed differentiation protocol to generate ventricular-like CMs from iPSCs derived from 3 healthy individuals[31]. On day 15, iPSC-CMs were digested and distributed to 4 experimental groups (Fig. 1a). B27 medium, routinely used for iPSC-CM culture, served as a control. To unravel the synergistic effects of MM, NP, and ES on the maturation of iPSC-CMs, we systematically applied NP and ES to MM in a stepwise parallel manner (Fig. 1a). MM was designed based on a published FA-supplemented medium that enhances the metabolic maturation of iPSC-CMs[13], with some modifications (Supplementary Table 1). NP was used to induce cell alignment and ES was applied to induce a beating frequency of 2 Hz (Supplementary Movie 1).

## Combined approach enhances structural maturation of iPSC-CMs
We observed that NP application induced changes in cell shape and a significant increase in the alignment of iPSC-CMs in the MM + NP and MM + NP + ES groups compared to the B27 and MM groups (Fig. 1b). Striated pattern of sarcomeres, as demonstrated by the immunostaining against the sarcomeric protein α-actinin, was observed in iPSC-CMs under all conditions (Fig. 1c). Notably, highly organised sarcomeres with well-defined striations, which are aligned into long, continuous myofibrils (stained with phalloidin) that run the length of the cell, were observed only in the MM + NP and MM + NP + ES groups (Fig. 1c, d). Quantitative analysis revealed that the majority of iPSC-CMs in the MM + NP and MM + NP + ES groups showed sarcomere patterns at an angle of 90° to the NP direction with elongated nuclei along the NP direction, whereas iPSC-CMs in the B27 and MM groups revealed random orientations of sarcomeres and nuclei in all directions (Fig. 1c, e, f). In addition, the sarcomere length is significantly longer in iPSC-CMs in the MM + NP and MM + NP + ES groups compared to those in the B27 and MM groups (Supplementary Fig. 1b). Co-immunostaining for α-actinin and cardiac ryanodine receptor (RYR2) revealed that, in B27-cultured CMs, robust RYR2 staining was observed as punctate staining with a low degree of α-actinin/RYR2 colocalisation. In contrast, CMs from the other three groups revealed an augmented presence of striated patterns. The α-actinin/RYR2 colocalisation was significantly enhanced in the MM + NP + ES group compared to the B27 and MM groups (Fig. 1c, Supplementary Fig. 1c). In the B27 and MM groups, the gap junction protein connexin 43 (Cx43) was partially localised to perinuclear regions and in the cytosol, whereas Cx43 membrane localisation was increased in CMs of the MM + NP and MM + NP + ES groups (Fig. 1d). These results provide evidence for the additive effects of NP and ES to MM on the structural maturation of iPSC-CMs.

## Combined approach improves electrophysiological maturation
To evaluate the effect of MM, NP and ES on electrophysiological properties, we performed patch clamp and multi-electrode array (MEA) studies to investigate the action potential (AP) and field potential (FP) parameters of single and monolayer iPSC-CMs, respectively. We observed 43% of iPSC-CMs in the MM + NP + ES group with a 'notch-and-dome' AP morphology, which was not seen in the other groups (Fig. 2a). The resting membrane potential (RMP) was found to be progressively more negative in CMs from the MM ($-49.7 \pm 8.5$ mV), MM + NP ($-58.2 \pm 7.4$ mV) and MM + NP + ES ($-65.6 \pm 8.5$ mV) groups compared to the B27 group ($-44.1 \pm 9.8$ mV), while the maximum AP upstroke velocity (Vmax) was gradually increased in iPSC-CMs from $4.2 \pm 1.4$ V/s (B27) to $5.0 \pm 1.1$ V/s (MM), $6.6 \pm 2.5$ V/s (MM + NP) and $11.0 \pm 7.4$ V/s (MM + NP + ES). Similarly, a gradual increase in AP amplitude (APA) was observed in the four groups (Fig. 2b, Supplementary Table 2). The AP duration at 90% repolarisation ($APD_{90}$) was significantly shorter in iPSC-CMs paced at 0.5 Hz in the MM + NP + ES group than in the B27 control (Fig. 2c). As the transient outward $K^+$ current ($I_{to}$) underlies the prominent phase 1 repolarisation of cardiac APs and the 'notch-and-dome' AP morphology, we measured $I_{to}$ and found a significantly higher $I_{to}$ density in iPSC-CMs from the MM, MM + NP and MM + NP + ES groups compared to the B27 group, particularly in MM + NP + ES; and NP itself has less effect on $I_{to}$ when comparing the MM + NP group with the MM group (Fig. 2d, e).

Intercellular electrotonic coupling and conduction velocity (CV) across CMs are largely dependent on Cx43 expression at the gap junction[32]. Consistent with the increased expression of Cx43 on the cell membrane (Fig. 1d), heatmaps of electrical signal propagation analysed using MEA illustrate the stepwise increase in CV in iPSC-CMs in the MM ($22.3 \pm 3.7$ cm/s), MM + NP ($25.6 \pm 4.3$ cm/s) and MM + NP + ES ($27.8 \pm 7.3$ cm/s) groups, compared to the B27 condition ($12.5 \pm 5.8$ cm/s). Similar stepwise changes in spike amplitude and

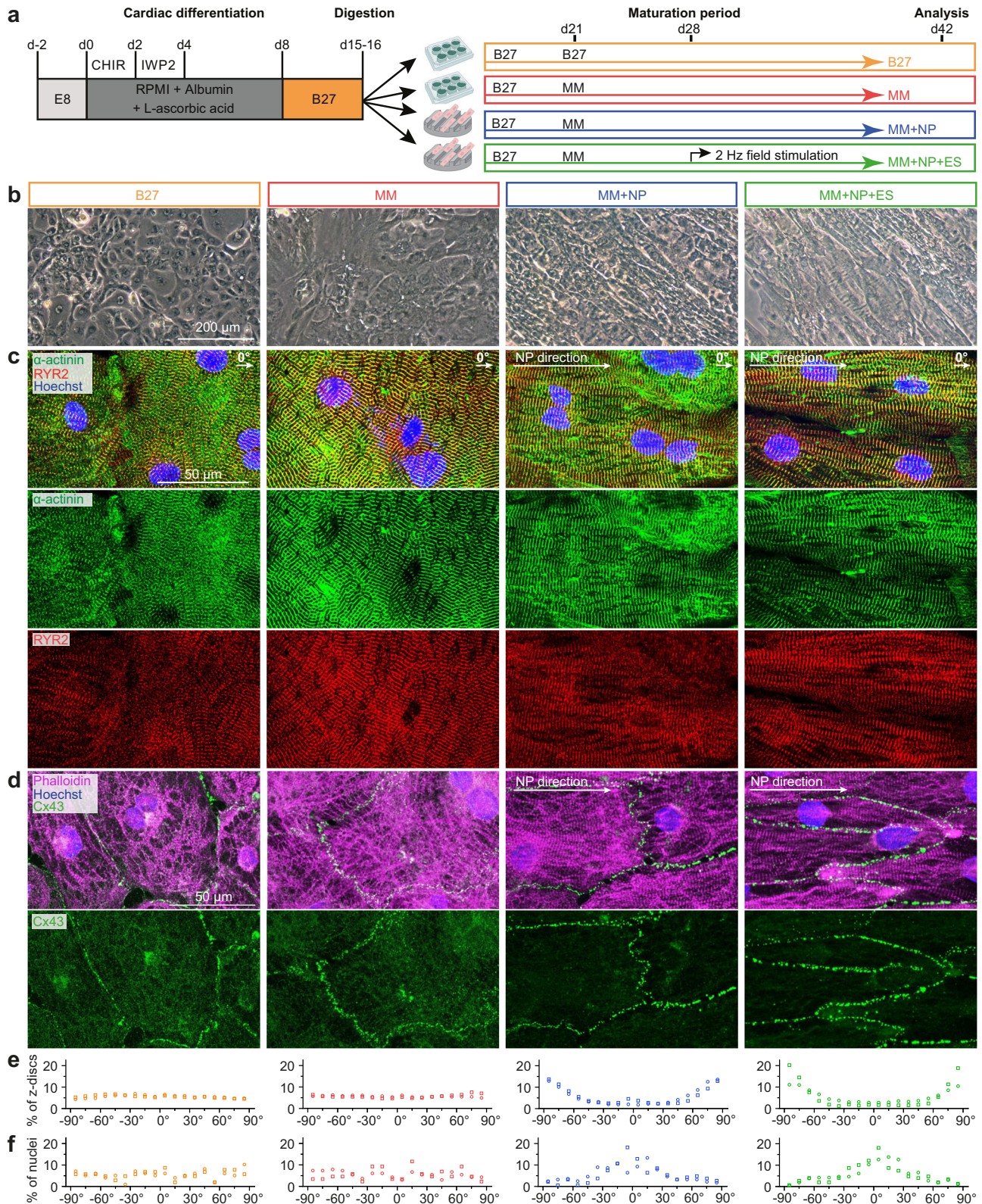

slope were observed in the four groups (Fig. 2f, g, Supplementary Table 3).

To analyse which changes in specific ion currents underlie the improved electrophysiological functionality, we recorded $I_{Na}$, $I_{K1}$, and $I_{Kr}$ using the patch clamp technique. We found that $I_{Na}$ density was significantly higher in MM-cultured iPSC-CMs with a mean peak current density of $-81.6 \pm 6.3$ pA/pF at $-25$ mV compared to the B27 group

($-42.3 \pm 4.7$ pA/pF at $-20$ mV). Notably, NP induced only a small increase in $I_{Na}$ density with a mean peak of $-92.1 \pm 6.9$ pA/pF when compared to the MM group, but $I_{Na}$ density was further significantly induced by ES in the MM + NP + ES group with a mean peak of $-135.7 \pm 9.8$ pA/pF (Fig. 3a, b). These data are consistent with the AP-(Vmax, APA) and FP-metrics (CV, spike amplitude and slope) shown in Fig. 2.

**Fig. 1 | Study design and structural characterisation of iPSC-CMs. a** Schematic overview of the study design. Differentiated iPSC-CMs were digested on day 15-16 (d15-16) and randomly divided into 4 experimental groups to investigate the effects of maturation medium (MM), nanopatterning (NP) and electrostimulation (ES). Extensive characterisation of the cells was performed on day 42 (d42). For some experiments, including calcium imaging, seahorse assays and multi-electrode array measurements, iPSC-CMs were replated into the corresponding assay plates on d42 and allowed to recover for another 7 days. Created in BioRender. Li, W. (2025) https://BioRender.com/p85e700. **b** Representative morphology of iPSC-CMs under different conditions at d42. Scale bar, 200 μm for all four groups. **c** Representative immunostaining for α-actinin, cardiac ryanodine receptor (RYR2) and Hoechst33342. **d** Representative immunostaining for connexin 43 (Cx43), phalloidin and Hoechst33342. Data (**b-d**) are based on 3 independent experiments using 3 different iPSC lines. **e, f** Quantification of sarcomere alignment based on z-disk orientation (**e**) and nuclei elongation (**f**). The method used to analyse the sarcomere alignment is shown in Supplementary Fig. 1a. Data were obtained from 2 independent experiments using 2 iPSC lines (*n* = 4 images/group/experiment). Data are presented as averages of each experiment. Orange, red, blue and green bars represent the four experimental groups: B27, MM, MM + NP, and MM + NP + ES, respectively. Symbols denote iPSC lines: circles for isWT7, and squares for iWTD2. Source data are provided as a Source Data file.

The electrophysiological immaturity of iPSC-CMs compared to adult CMs is partly attributed to the low density of the hyperpolarising K$^+$ current $I_{K1}$, which is important for stabilising the RMP[33]. We found a very low $I_{K1}$ density in B27-cultured iPSC-CMs ($-2.1 \pm 0.3$ pA/pF at $-130$ mV), and only a slight increase in $I_{K1}$ in the MM group ($-5.3 \pm 0.8$ pA/pF). Interestingly, NP induced a significant increase in $I_{K1}$ in the MM + NP group ($-11.7 \pm 1.8$ pA/pF), which was further induced by ES ($-16.0 \pm 1.9$ pA/pF in the MM + NP + ES group) (Fig. 3c, d). HERG channels conducting the rapid delayed rectifier K$^+$ current $I_{Kr}$ are involved in phase 3 repolarisation of cardiac APs. Similar to $I_{K1}$, both E-4031-sensitive $I_{Kr}$ step and tail current densities were only slightly induced by MM, but significantly induced by NP and further enhanced by ES (Fig. 3e-g). These data are consistent with the most negative RMP and the shortest APD$_{90}$ in iPSC-CMs from the MM + NP + ES group (Fig. 2b, c).

Taken together, these findings highlight the distinct effects of the three stimuli on specific ion currents. This is evidenced by the strong influence of NP on $I_{K1}$ and $I_{Kr}$, with less or no effect on $I_{Na}$ and $I_{to}$, and the robust effect of MM on $I_{Na}$, but less on $I_{Kr}$. Importantly, the data underline that the combined approach significantly enhanced electrophysiological maturation of iPSC-CMs.

## Combined approach improves calcium handling and contractility

Since excitation-contraction coupling in CMs involves calcium cycling to convert electrical signals into mechanical output (contraction), we next examined L-type calcium channel (LTCC) current $I_{Ca-L}$ and calcium transients in iPSC-CMs. $I_{Ca-L}$ densities exhibited similar reductions in both the MM and MM + NP groups when compared to the B27 group, which were further reduced by ES (Fig. 4a, b). To quantify intracellular Ca$^{2+}$ dynamics, we performed Fura-2-based calcium imaging in iPSC-CMs paced at 0.5 Hz (Fig. 4c-e). Significantly reduced diastolic and systolic Ca$^{2+}$ levels were observed in the MM, MM + NP and MM + NP + ES groups compared to the B27 group, but Ca$^{2+}$ transient amplitudes were comparable between all groups despite the reduced $I_{Ca-L}$ density in the MM, MM + NP and MM + NP + ES groups. The Ca$^{2+}$ transient decay time constant (tau) is also significantly shortened in the MM group compared to the control, whereas no further shortening was observed in the MM + NP and MM + NP + ES groups. Application of 10 mM caffeine resulted in significantly increased Ca$^{2+}$ release from the sarcoplasmic reticulum (SR) in the MM, MM + NP and MM + NP + ES groups compared to the B27 group (Fig. 4e). These data suggest a more efficient coupling between $I_{Ca-L}$ and SR Ca$^{2+}$ release, enhanced Ca$^{2+}$ decay kinetics and a higher SR calcium content in these three groups.

Previous studies reported that high calcium concentration induces positive force-frequency behaviour, physiological twitch kinetics and robust β-adrenergic response of EHT by improving Ca$^{2+}$ handling[30]. To investigate the contribution of the high calcium concentration in MM to the maturation of iPSC-CMs, we cultured iPSC-CMs in DMEM + B27 medium containing 1.8 mM calcium (Supplementary Table 1) compared to (RPMI + )B27 medium containing 0.4 mM calcium (Fig. 4f-j). Analysis of the calcium transient parameters revealed only slightly increased Ca$^{2+}$ transient amplitudes but significantly decreased tau in iPSC-CMs cultured in DMEM + B27 compared to (RPMI + )B27 (Fig. 4g). Furthermore, caffeine-induced SR Ca$^{2+}$ release was significantly increased in the DMEM + B27 group compared to the (RPMI + )B27 group (Fig. 4h). Interestingly, all changes were similar to those in the MM group (Fig. 4d, e). We also found that cultivation in DMEM + B27 led to an increase in $I_{Na}$ density to $-90.4 \pm 7.2$ pA/pF at $-20$ mV compared to $-42.3 \pm 4.7$ pA/pF in the RPMI + B27 group (Fig. 4i). $I_{Ca-L}$ density was significantly reduced in the DMEM + B27 group ($-6.2 \pm 0.8$ pA/pF at 10 mV) compared to the B27 group ($-8.4 \pm 0.5$ pA/pF) (Fig. 4j). Both $I_{Na}$ and $I_{Ca-L}$ in the DMEM + B27 group are comparable to those in the MM group (Figs. 3a and Fig. 4b). Overall, these results demonstrate the strong influence of high Ca$^{2+}$ concentrations on the electrophysiological properties of iPSC-CMs.

Movie-based analysis of iPSC-CM beating properties[5] showed that the changes in calcium handling were associated with improved contractile function. Stopping ES in the MM + NP + ES group resulted in cessation of beating, followed by regaining of spontaneous beating activity within 15-30 minutes at a rate comparable to the other three groups (Fig. 5a). Significantly shorter contraction and relaxation times and beating duration were observed in the MM + NP + ES group compared to the other groups (Fig. 5b; Supplementary Fig. 2a, b). We found a similar trend in the MM and MM + NP groups compared to the B27 control, but no significant difference between the two groups. Consistent with this observation, all cultures in the MM + NP + ES group successfully captured high-frequency (2 Hz) field stimulation, whereas none of the B27 cultures demonstrated this capability (Fig. 5c; Supplementary Fig. 2c). This improved contractile function was accompanied by an increased gene expression ratio of *TNNI3/TNNI1* and *MYL2/MYL7*, whereas the expression of *MYH6*, encoding the fast-twitch MHC isoform, was upregulated in response to sustained ES, leading to a reduced *MYH7/MYH6* ratio (Fig. 5d). Flow cytometry analysis showed that in all three groups (MM, MM + NP, and MM + NP + ES) there was a noticeable increase in cell volume and granularity of iPSC-CMs compared to the B27 control. Compared to the MM group, NP did not induce an additional increase in cell volume and granularity, but the combination of MM + NP + ES induced further significant increases (Fig. 5e), suggesting that ES plays an important role in the hypertrophic growth of iPSC-CMs. Whereas a comparable proportion of cardiac troponin T (cTNT)-positive cells was found in all conditions, the highest cTNT mean fluorescence intensity was observed in iPSC-CMs from the MM + NP + ES group (Fig. 5f, g). These results demonstrate that MM, NP and ES individually and synergistically induce electrophysiological and functional maturation of iPSC-CMs.

## Combined approach improves drug response of iPSC-CMs

To investigate whether the maturation state of iPSC-CMs influences their drug response, we chose verapamil (calcium-channel blocker), E-4031 (hERG-channel blocker) and isoprenaline (β-adrenergic stimulus) as model substances to detect pro-arrhythmic activity based on changes in FP parameters (Fig. 6; Supplementary Fig. 4a, b). We observed beating arrest in cultures from the B27 (17/17), MM (6/17) and MM + NP (6/18) groups at 1 μM verapamil, but no significant induction

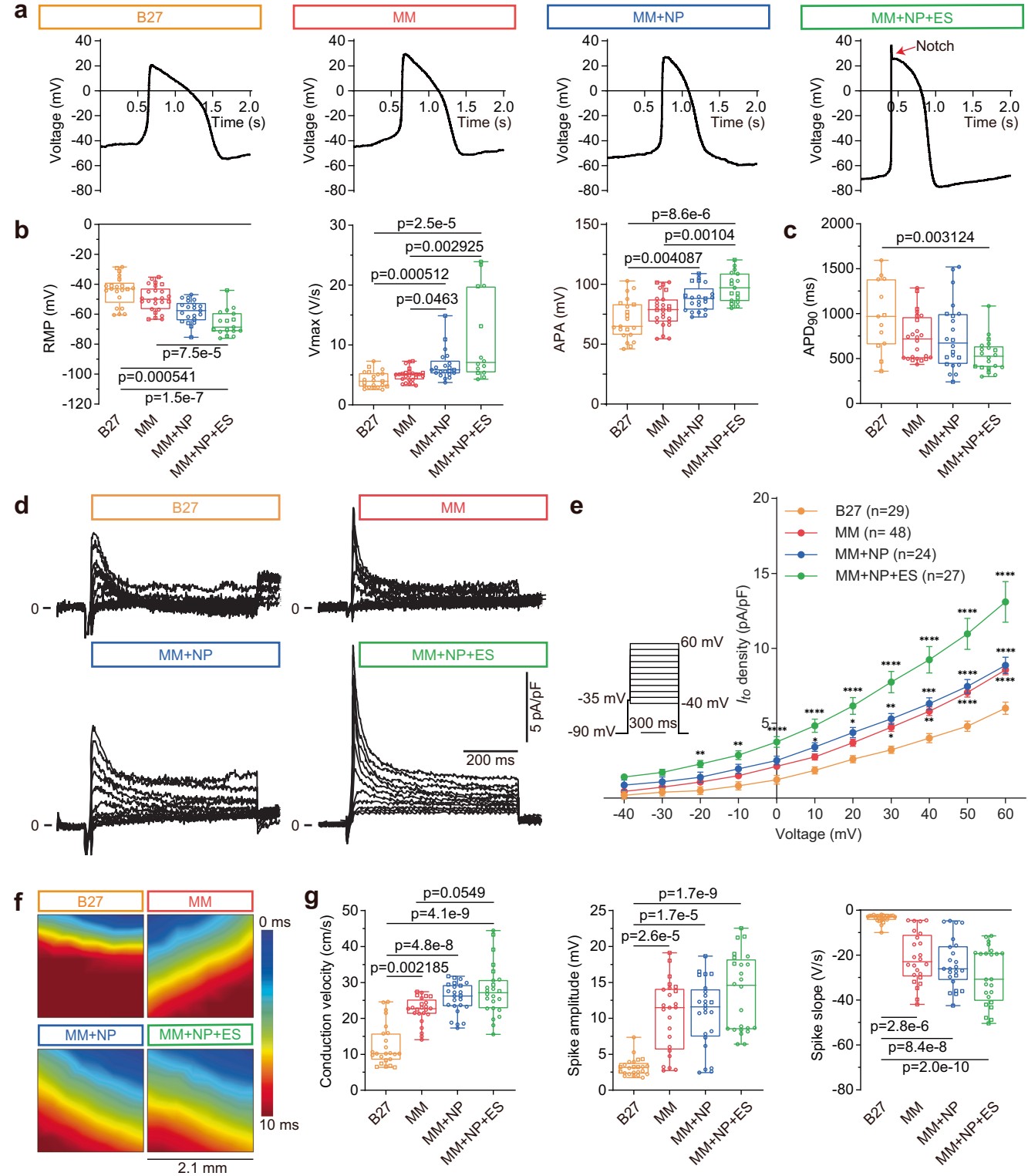

of arrhythmias with all three substances (Supplementary Fig. 4c, d). Concentration-dependent reductions in spike amplitude were observed for verapamil in the B27 group and, to a lesser extent, in the MM and MM + NP groups. In contrast, no effect of verapamil on beating activity and spike amplitude was observed in the MM + NP + ES group (Fig. 6b; Supplementary Fig. 4e). Verapamil-induced shortening of FP duration (FPDc), which corresponds to QT-shortening in the clinic, was comparable in iPSC-CMs from the MM, MM + NP and MM + NP + ES groups. However, this effect was more pronounced

compared to iPSC-CMs cultured in B27 (Fig. 6c, d). Previous studies showed that immature iPSC-CMs failed to produce APD prolongation after E-4031 treatment, even at high concentrations[34]. Similarly, we found that E-4031 induced only minor changes in FPDc in B27-cultured iPSC-CMs, whereas significant concentration-dependent FPDc prolongation was detected in the MM, MM + NP and MM + NP + ES groups (Fig. 6e, f). Furthermore, we observed a more pronounced positive-chronotropic response of iPSC-CMs to isoprenaline in these three groups than in the B27 group, which correlates with FPDc-shortening

**Fig. 2 | Assessment of action potential and field potential parameters in iPSC-CMs. a** Spontaneous action potential (AP) traces from the four groups. A notch event (red arrow) is only present in the MM + NP + ES group. **b** Quantification of spontaneous AP metrics: resting membrane potential (RMP), maximum upstroke velocity (Vmax), and AP amplitude (APA). $n = 21, 25, 21$ and 14 (Vmax) or 17 (RMP and APA) cells from 4 independent differentiations for the four groups, respectively. **c** $APD_{90}$ quantification in APs paced at 0.5 Hz ($n = 13, 24, 22$ and 20 cells from 4 independent differentiations for the four groups, respectively). **d** Representative $I_{to}$ traces recorded in iPSC-CMs from the four groups. **e** Statistical analysis of $I_{to}$ in single cells derived from 7 (B27), 9 (MM), and 6 (MM + NP, MM + NP + ES) independent differentiations of 3 iPSC lines. The stimulation protocol is shown as an inset. The holding potential was set at −90 mV. To inactivate $I_{Na}$, a 20-ms pre-pulse to −35 mV was applied. $I_{to}$ was recorded by increasing the test potential from −40 mV to +60 mV in 10 mV increments, with each pulse lasting 600 ms. **f** Representative heatmaps of field potential (FP) propagation. The colour scale

indicates the time at which the sodium peak of the FP signal propagates from the start time (0 ms) to end points (10 ms). Scale bar indicates the distance of the sodium peak propagation. **g** Quantification of FP parameters: conduction velocity, spike amplitude and spike slope. $n = 24$ cultures/group from 4 independent differentiations of 2 iPSC lines. A schematic diagram of FP traces is included in Fig. 6a, showing how the data for spike amplitude and slope were analysed. Symbols in (**b**, **c**, **g**) denote iPSC lines: circles for isWT7, and squares for iWTD2. Source data are provided as a Source Data file. Statistical analysis was performed using Kruskal-Wallis test with Dunn's multiple comparison test (**b**, **c**, **g**) and two-way ANOVA with Sidak's multiple comparison test (**e**): *$p < 0.05$, **$p < 0.01$, ***$p < 0.001$, ****$p < 0.0001$ compared to the B27 group. Exact $p$ values are provided in the Source Data file. Data are presented in box plots indicating median (middle line), 25th, 75th percentile (box) and min and max data points (whiskers) in (**b**, **c**, **g**) and in line plots as mean ± SEM (**e**).

(Fig. 6g, h). The $EC_{50}$ of isoprenaline for the chronotropic effect was also much lower in the MM + NP (1.9 nM) and MM + NP + ES (1.8 nM) groups than in the MM (4.3 nM) and B27 (6.8 nM) groups (Fig. 6h). These experiments demonstrate the substantial impact of the maturation state of iPSC-CMs on their response to various cardioactive drugs. They emphasise the importance of utilising iPSC-CMs with more adult-like electrophysiological properties for accurate drug risk assessment.

## Combined approach downregulates MAPK/PI3K-AKT pathways

Previous studies have shown that FA-enriched media induce iPSC-CM maturation by regulating key genes involved in FA metabolism, mitochondrial function, calcium cycling, ion channels and sarcomere[13,20]. To gain insight into the molecular mechanisms driving iPSC-CM maturation by NP and ES, we performed RNA sequencing (RNA-seq) analysis. Surprisingly, NP had little synergistic effect when combined with MM, whereas the addition of ES strongly influenced gene expression (Fig. 7a-d; Supplementary Fig. 5a, b). Comparing the MM and MM + NP groups, only 163 differentially expressed genes were identified, of which 56 were upregulated and 107 downregulated in the MM + NP group. In contrast, 1370 significantly upregulated and 1657 downregulated genes were identified in the MM + NP + ES group compared to the MM group, of which 747 significantly upregulated and 990 downregulated genes were also identified when compared to the MM + NP group, indicating the synergistic effects of NP and ES.

Pathway enrichment analysis of the downregulated genes in the MM + NP + ES group mainly mapped to MAPK/PI3K-AKT, TNFR2-NFκB, G-protein-coupled receptor (GPCR), and cytokine/chemokine signalling (Fig. 7e; Supplementary Fig. 5d; Supplementary Data 1). Using the transcription factor targets (TFT) collection (https://www.gsea-msigdb.org/gsea/msigdb/human/genesets.jsp?collection=TFT), we identified the SRF cluster, which includes many genes involved in cell cycle regulation and cell proliferation (Fig. 7f, g; Supplementary Fig. 5c). We also found a decrease in SRF protein levels in CMs from the MM + NP + ES group compared to the other groups (Fig. 7h). These findings encouraged us to evaluate the expression of genes that regulate cell cycle progression. Notably, activators of G2/M checkpoints including cyclins (*CCNB1-3*), cyclin-dependent kinase 1 (*CDK1*) were downregulated, whereas CDK inhibitors (*CDKN1A, CDC20*) were upregulated in the MM + NP + ES group compared to the other two groups (Fig. 8a, b). Interestingly, genes (*ANLN, SEPTIN7/2*) encoding activators of cytokinesis were also downregulated. We did not observe any significant changes in the gene sets (*CCND1-3, CCNE1/2, CCNA1/2, CDK2/4/6, CDKN2A-D, CDKN1B/C, CDH1*) controlling the G1 and S phase progression and the G1/S checkpoint (Fig. 8a, b). These data suggest a cell cycle arrest after S phase and before exit from M phase in the MM + NP + ES group, which may lead to bi-nucleation or nuclear polyploidy. To confirm this, we examined DNA content and found that the number of diploid iPSC-CMs was significantly reduced and

polyploid cells significantly increased in the MM + NP + ES group compared to the other three groups (Fig. 8c, f). Interestingly, we observed no difference in the proportion of 5-ethynyl-2'-deoxyuridine (EdU)-incorporated iPSC-CMs between all four groups, which can detect DNA synthesis during S phase (Fig. 8d, g). However, the proportion of Ki67+ iPSC-CMs and Ki67- polyploid iPSC-CMs was higher in the MM + NP + ES group than in the other groups (Fig. 8e, h, i). Ki67 is widely expressed throughout the entire cell cycle, except in G0, and reaches a maximum in S/G2[35]. These results indicate that the downregulation of MAPK/PI3K-AKT and SRF-related genes is involved in G2/M arrest and polyploidy development of iPSC-CMs.

## Gene expression profile has links to metabolism and electrophysiology

The pathway enrichment analysis of genes upregulated in the MM + NP + ES group revealed that the combined approach induced the upregulation of genes involved in electron transport chain (ETC), TCA cycle, mitochondrial biogenesis, NRF2 signalling, glucose metabolism, N-glycan biosynthesis, FA oxidation, tRNA aminoacylation, etc. (Fig. 9a; Supplementary Fig. 5e; Supplementary Data 2). These gene clusters were only slightly upregulated in the MM + NP group compared to the MM group (Supplementary Fig. 5e).

Using the TFT collection we identified two clusters, TFAM (Fig. 9b) and HMCES (Fig. 9c), which were enriched in the MM + NP + ES group (Supplementary Fig. 5c). In these two clusters, mitochondrial DNA (mtDNA)-encoded NADH dehydrogenase subunits (*MTND2-6*) and pseudogenes (*MTND4P12, MTND5P11, MTND6P4*), cytochrome c oxidase subunits (*MT-CO1/3*) and pseudogenes (*MT-CO1P2, MT-CO1P12, MT-CO3P12*), cytochrome b (*MTCYB*), 12S rRNA (*MT-RNR1*) and tRNAs (*MT-TP/-TM/-TE/-TI/-TW/-TC/-TY/-TR/-TN/-TQ*) were upregulated in the MM + NP + ES group compared to the MM group.

Subsequently, we found increased mitochondrial mass in CMs from the MM + NP + ES group compared to the other groups (Fig. 9d). Furthermore, CMs from the MM + NP + ES group showed a higher expression of *OPA1, PPARGC1α* and *PPARα* (Fig. 9g), confirming an enhanced mitochondrial development in response to ES. Measurements of oxygen consumption rate as a surrogate for mitochondrial function showed an increased basal and maximal respiration, ATP production and spare capacity of CMs from the MM group compared to the B27 control. There is only a marginal additional increase in these parameters in the MM + NP + ES and MM + NP groups compared to the MM group (Fig. 9e, f). These findings indicate that the combination of MM + NP + ES leads to upregulation of oxidative phosphorylation, activation of mtDNA-encoded components and an overall enhancement of mitochondrial development and function.

Further examination of individual changes in key ion channel components revealed a clear correlation between gene expression and channel function (Fig. 9h). We found upregulation of genes contributing to $I_{to}$ (*KCNA7, KCNC3, KCNC4, KCND2, KCND3*), $I_{K1}$ (*KCNJ2*,

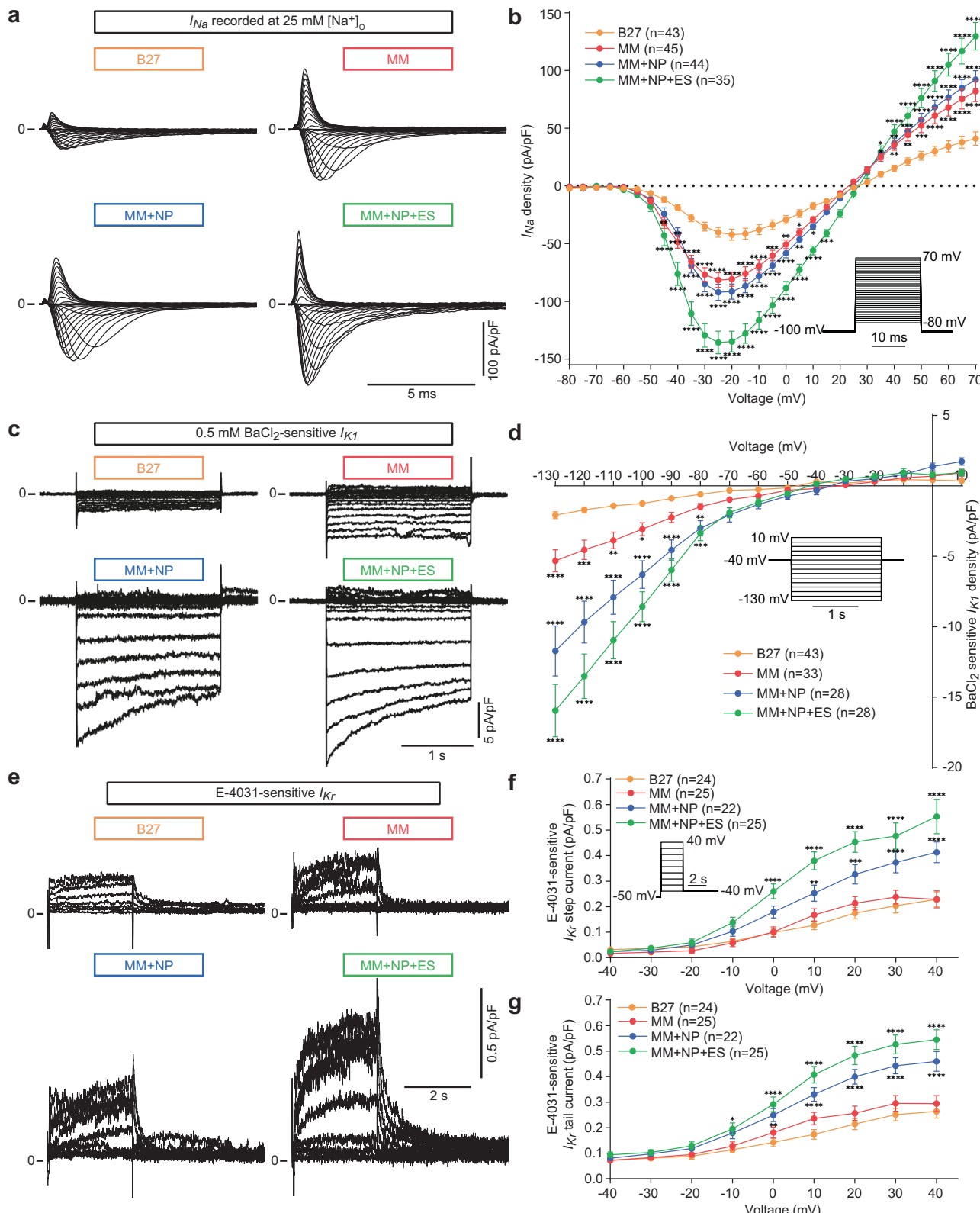

KCNJ4, KCNJ6, KCNJ11, KCNJ12), $I_{Kr}$ (KCNH2, KCNE2) and $I_{Ks}$ (KCNQ1) currents, as well as genes encoding calcium-activated (KCNN1) and voltage-gated (KCNS3, KCNAB2, KCNIP3) potassium channels in the MM + NP + ES group compared to the MM group. In contrast, genes encoding the LTCCs (CACNA1C, CACNA1D, CACNA1A, CACNA2D3, CACNA2D4) and regulating the LTCC activity (CACNG7) were down-regulated, whereas the genes encoding the T-type calcium channels

(CACNA1H, CACNA1I) were upregulated. No significant differences in SCN5A expression were found between the three groups, but SCN1B was upregulated and SCN3B was downregulated in the MM + NP + ES group. In addition, the expression of the genes encoding calcium handling proteins were not significantly altered. Collectively, these results are consistent with the significantly increased $I_{Na}$, $I_{to}$, $I_{K1}$, and $I_{Kr}$ but decreased $I_{Ca-L}$ currents in iPSC-CMs from the MM + NP + ES group.

**Fig. 3 | $I_{Na}$, $I_{K1}$ and $I_{Kr}$ recordings in iPSC-CMs. a** Representative $I_{Na}$ traces recorded under 25 mM extracellular $Na^+$ concentration. **b** Statistical analysis of $I_{Na}$ in single cells derived from 9 (B27) and 5 (MM, MM + NP, MM + NP + ES) independent differentiations of 3 iPSC lines. The stimulation protocol is shown as an inset. The holding potential was set at −100 mV. $I_{Na}$ was recorded by increasing the test potential from −80 mV to +70 mV in 5 mV steps. Each pulse lasted for 20 ms and the sweep interval was 2 s. **c** Representative traces of 0.5 mM $BaCl_2$-sensitive $I_{K1}$ for different groups. **d** Statistical analysis of $BaCl_2$-sensitive $I_{K1}$ in single cells derived from 6 (B27, MM, MM + NP + ES) and 5 (MM + NP) independent differentiations of three iPSC lines. The stimulation protocol is shown as an inset. The holding potential was set at −40 mV. $I_{K1}$ was recorded by increasing the test potential from −130 mV to +10 mV in 10 mV steps, with each pulse lasting for 2 s. The sweep interval was 10 s. The protocol was then repeated in the presence of 0.5 mM $BaCl_2$

and the $Ba^{2+}$-sensitive current was calculated as $I_{K1}$. **e** Shown are traces of 1 μM E−4031-sensitive $I_{Kr}$ in the four groups. **f, g** Averaged E−4031-sensitive $I_{Kr}$ step currents (**f**) and tail currents (**g**) in single cells derived from 4 independent differentiations of 2 iPSC lines. The $I_{Kr}$ pulse stimulation is shown as an inset. The holding potential was set at −50 mV. $I_{Kr}$ step currents were recorded by decreasing the test potential from +40 mV to −40 mV in 10 mV steps with each pulse lasting for 2.5 s. The tail currents were recorded in the following 4-s phase at −40 mV. The sweep interval was 10 s. With respect to $I_{Na}$, $I_{K1}$, and $I_{Kr}$, we did not observe significant differences among three iPSC lines used (Supplementary Fig. 7). Source data are provided as a Source Data file. Two-way ANOVA with Sidak's multiple comparison test was used for statistical analysis: *$p < 0.05$, **$p < 0.01$, ***$p < 0.001$, ****$p < 0.0001$ compared to the B27 group. Exact $p$ values are provided in the Source Data file. Data are presented as mean ± SEM.

## MM + ES alone induces electrophysiological maturation and polyploidy of iPSC-CMs

As MM + NP had little synergistic effect on gene expression profile when compared to MM alone, it is interesting to test whether MM + ES alone can induce the maturation of iPSC-CMs similar to MM + NP + ES. To study this, we compared iPSC-CMs cultured under MM and MM + ES conditions. We observed no significant differences in cell morphology, cTNT intensity, and cell size (FSC-A) (Fig. 10a-c). Consistent with the changes observed in the MM + NP + ES group, iPSC-CMs in the MM + ES group showed increased granularity (SSC-A), and Tom20 intensity (Fig. 10c, d). These data indicate that ES alone enhances structural maturation, whereas the addition of NP may contribute to cell alignment, cell size and cTNT intensity. We observed the 'notch-and-dome' AP morphology in iPSC-CMs (32%) of the MM + ES group (Fig. 10e) comparable to those in the MM + NP + ES group. iPSC-CMs in the MM + ES group had a more negative RMP, increased Vmax and APA, and shortened $APD_{90}$ compared to the MM group (Fig. 10f), but with a less extent when compared to the MM + NP + ES group (Fig. 2b). Similarly, $I_{to}$ density was increased in the MM + ES group compared to the MM group (Fig. 10g). Investigation of the contractile function revealed a slight shortening of the contraction time, relaxation time and beating duration of iPSC-CMs in the MM + ES group compared to MM (Fig. 10h), but these changes were smaller compared to those observed with MM + NP + ES versus MM (Fig. 5a). Similar to the MM + NP + ES group, all iPSC-CM cultures in the MM + ES group captured the pacing frequency of 2 Hz (Fig. 10i).

To establish whether ES is the pivotal stimulus for the development of polyploidy of iPSC-CMs, DNA content was measured in iPSC-CMs. The proportion of diploid iPSC-CMs was significantly reduced, while the number of polyploid CMs was significantly increased in the MM + ES group compared to the MM group (Fig. 10j, l). Additionally, we observed the increase in the number of Ki67+ and Ki67- polyploid iPSC-CMs in the MM + ES group compared to the MM group (Fig. 10k, m, n), similar to those in the MM + NP + ES group (Fig. 8h, i). Taken together, these results demonstrate that ES alone can enhance mitochondrial development, electrophysiological function and polyploidy of iPSC-CMs.

## Discussion

In this study, we systematically investigated the collective impact of MM, elevated calcium concentration in the medium, NP and ES on the maturation of iPSC-CMs. Our findings demonstrate that the concurrent application of MM with a high concentration of calcium, NP and ES serves as an efficient strategy to enhance the structural, electrophysiological, metabolic and functional maturation of iPSC-CMs. This maturation process involves the modulation of MAPK/PI3K-AKT signalling as well as the regulation of TFAM/HMCES and SRF target genes, which is of significant relevance for the utilisation of iPSC-CMs in disease modelling and drug testing.

The core of the maturation strategy is the application of FA-supplemented MM in iPSC-CMs, which has been reported in several studies[13,16,20,22]. Previous studies have shown that MM induces the metabolic transition from glucose-based energy production to FA β-oxidation[13,20] and changes in the expression of genes associated with calcium cycling, ion channels and structural proteins[13,16,22]. This metabolic transition is essential for increased $Ca^{2+}$ transient kinetics, $I_{K1}$ density, and iPSC-CM hypertrophy and improved AP parameters, mitochondrial density and function, contractility and drug response not only in 2D-cultures[13,20] but also in 3D-tissues generated with iPSC-CMs and fibroblasts[16,22]. In line with these studies, we show here that MM induces enhanced structural (cell size, cTNT and Tom20 intensity), electrophysiological (increased $I_{Na}$, $I_{to}$, $I_{K1}$ and $I_{Kr}$ density, improved FP parameters and $Ca^{2+}$ cycling) and metabolic (mitochondrial respiration) maturation of iPSC-CMs, leading to improved contractility and drug response. The effect of MM on $I_{Na}$, $I_{Ca-L}$ and $Ca^{2+}$ cycling is largely due to the high concentration of calcium in MM.

One of the contributions of the NP surface to the maturation of iPSC-CMs is the improvement of cell alignment, granularity, sarcomere length and contractile behaviour, but their structural maturation is still less pronounced compared to 3D tissue. 3D tissue allows symmetric contractions of EHTs with a significantly improved structure, including T-tubule formation and colocalisation with RYR2, sarcomere organisation and length[16–18,28]. This difference may be due to the need of 2D monolayers to attach to the surface, where there is a significant mechanical mismatch. The NP surface with a Young's modulus of ~7 mPa is much higher compared to diastolic adult human myocardium in the range of 8-15 kPa[36] and is therefore still a major limitation for its use in 2D culture when compared to 3D tissue or 2D cultures with other techniques (e.g., on PDMS with 8 kPa)[24] that allow consistent fractional shortening across the tissue.

By stepwise addition of NP and ES to MM, we found that the combination of NP with MM had only a limited impact on the metabolic maturation of iPSC-CMs, as demonstrated by subtle changes in the expression profile of gene clusters related to energy metabolism as well as mitochondrial development and function, such as ETC (also known as oxidative phosphorylation), TCA cycle, FA oxidation, and glucose metabolism (Supplementary Fig. 5e). Strikingly, the addition of ES to MM + NP resulted in significant upregulation of these gene clusters, likely due to the increased FA β-oxidation in iPSC-CMs to meet the energy demand for the persistent beating activity at 2 Hz.

An important finding of our study is that GSEA analysis of the upregulated genes in MM + NP + ES using the transcription factor target database maps to the enrichment of HMCES- and TFAM-related target gene sets. HMCES (5-hydroxymethylcytosine binding, embryonic stem cell-specific) may safeguard the genomic and mtDNA integrity of iPSC-CMs during oxidative stress responses triggered by elevated levels of reactive oxygen species due to the use of FA. This protective mechanism involves the formation of stable DNA-protein crosslinks with abasic DNA damage to prevent error-prone repair pathways[37–39].

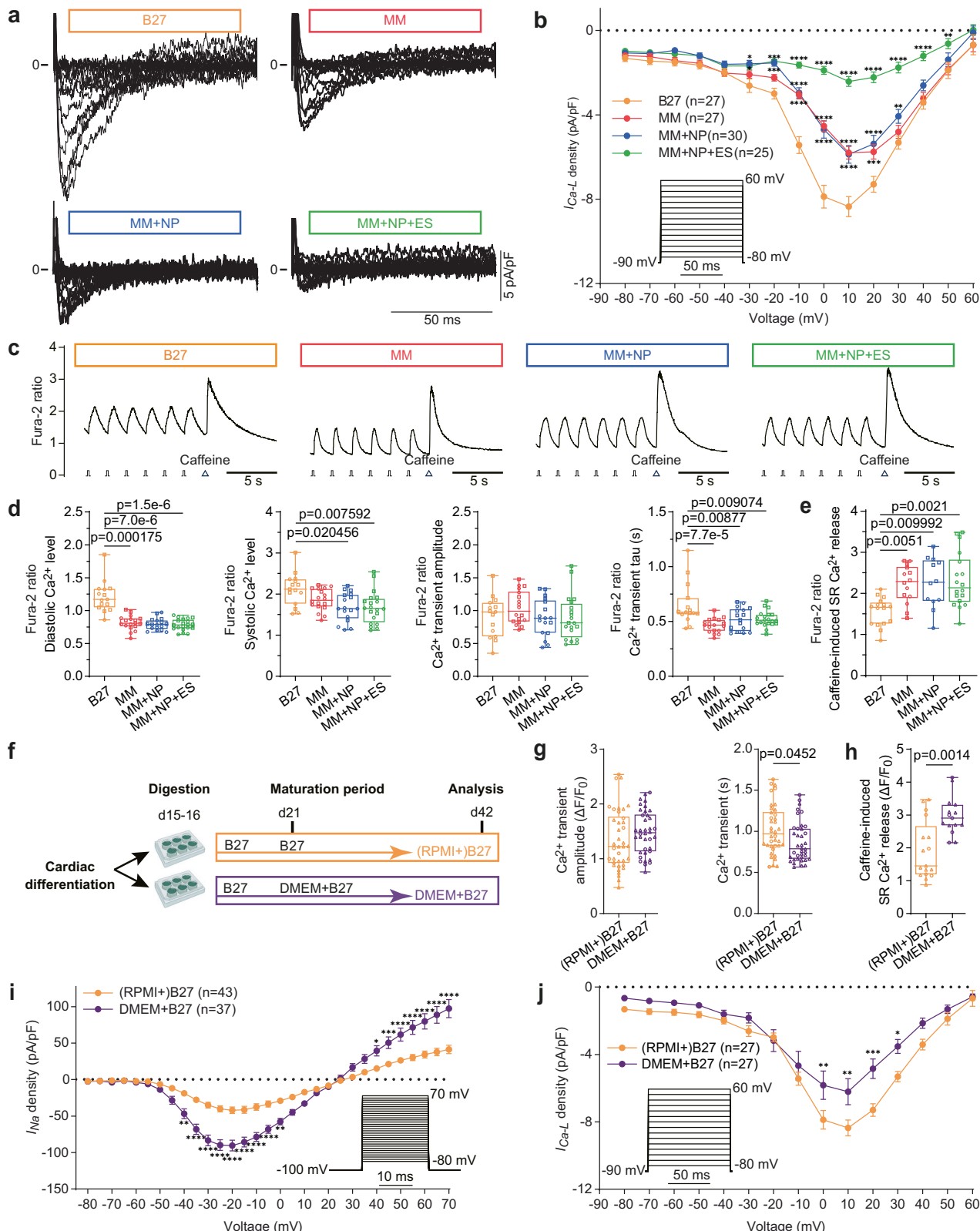

TFAM (mitochondrial transcription factor A) is essential for the transcription, replication and packaging of mtDNA into nucleoids. It is indispensable for the meticulous regulation of mitochondrial biogenesis, ensuring the seamless adaptation of the mitochondrial population to precisely match the energy demand of the cell[40]. This is supported by our observation that most mtDNA-encoded essential components of the ETC as well as rRNAs and tRNAs required for the translation of mtDNA-encoded proteins were significantly upregulated by the ES stimulation (Fig. 9).

PGC-1α (*PPARGC1α*) is the master regulator of mitochondrial energy metabolism, respiration and biogenesis through interaction with its various coactivators ERR, PPAR and NRF1/2[41]. Together with ERR, it controls mitochondrial dynamics via activation of genes involved in mitochondrial fission and fusion including *OPA1* and

**Fig. 4 | Assessment of calcium handling in iPSC-CMs and effects of calcium concentrations used in media. a** Representative $I_{Ca-L}$ recordings in iPSC-CMs cultured under the four conditions. **b** Statistical analysis of $I_{Ca-L}$ in single cells derived from 6 independent differentiations of 3 iPSC lines. **c** Representative $Ca^{2+}$ transient traces recorded at 0.5 Hz followed by application of 10 mM caffeine to induce the release of total SR calcium. **d** Statistical analysis of calcium transient parameters: diastolic $Ca^{2+}$ level, systolic $Ca^{2+}$ level, transient amplitude and decay time constant tau measured in iPSC-CMs paced at 0.5 Hz. $n = 15, 18, 17$, and 20 cells for B27, MM, MM + NP, and MM + NP + ES groups, respectively, from 2 iPSC lines. **e** SR calcium release induced by 10 mM caffeine. $n = 16, 14, 12$, and 18 cells for B27, MM, MM + NP, and MM + NP + ES groups, respectively, from 2 iPSC lines. **f–h** Experimental scheme (**f**), and effects of calcium concentrations on $Ca^{2+}$ transient amplitude and tau (**g**, $n = 38$ cells/group from 2 iPSC lines), and SR $Ca^{2+}$ release induced by 10 mM caffeine (**h**, $n = 18$ and 15 cells for the (RPMI + )B27 and

DMEM + B27 groups from 2 iPSC lines, respectively). **f** is created in BioRender. Li, W. (2025) https://BioRender.com/t88d878. **i** Effects of calcium concentrations on $I_{Na}$ in single cells derived from 9 ((RPMI + )B27) and 6 (DMEM + B27) independent differentiations of 3 iPSC lines. **j** Effects of calcium concentrations on $I_{Ca-L}$ in single cells derived from 6 ((RPMI + )B27) and 4 (DMEM + B27) independent differentiations of 3 iPSC lines. Symbols in (**d, e, g, h**) denote iPSC lines: circles for isWT7, squares for iWTD2 and triangles for iBM76. Source data are provided as a Source Data file. Statistical analysis was performed using Kruskal-Wallis test with Dunn's multiple comparison test (**d, e**) or two-sided Kolmogorov-Smirnov test (**g, h**). Two-way ANOVA with Sidak's multiple comparison test was used in (**b, i, j**) *$p < 0.05$, **$p < 0.01$, ***$p < 0.001$, ****$p < 0.0001$ compared to the B27 group. Exact $p$ values are provided in the Source Data file. Data are presented as mean ± SEM (**b, i, j**) and in box plots indicating median (middle line), 25th, 75th percentile (box) and min and max data points (whiskers) in (**d, e, g, h**).

$MFN2$[42], which were found to be upregulated in the MM + NP + ES group. Together with PPARα, which is also upregulated in the MM + NP + ES, PGC-1α regulates genes involved in mitochondrial FA oxidation and many other cellular lipid metabolic pathways[40,41]. Furthermore, PGC-1α together with NRF1/2 promotes mitochondrial biogenesis by activating TFAM[40,41].

Another interesting discovery of our study revealed a significant downregulation of genes involved in MAPK/PI3K signalling pathways in the MM + NP + ES group. This aligns with the substantial downregulation of the MAPK and PI3K-AKT pathways in the postnatal heart when compared to the neonatal heart[43]. Thus, MAPK/PI3K-AKT inhibition promotes iPSC-CM maturation, partially mediated by the upregulation of PGC-1α[43]. Further research is needed to determine whether the downregulation of MAPK/PI3K signalling pathways in the MM + NP + ES group is involved in PGC-1α and TFAM activation through the control of AMPK[41,44], AKT[45] and/or mTORC1[46] pathways, and is therefore essential for the metabolic maturation of iPSC-CMs.

Most strikingly, our RNA-seq and cell cycle analysis data show that downregulated MAPK/PI3K-AKT signalling is involved in the downregulation of genes important for the G2/M transition and cytokinesis, leading to polyploidy (nuclear polyploidy and/or multinucleation) of iPSC-CMs. Human CMs are diploid during the first years of life and gradually become polyploid over time. By the second decade of life, approximately 25% of CM are multinucleated and 57% polyploid[47]. In CMs, the major cyclin-CDK complexes controlling cell cycle progression are CCND-CDK4/6 (G1 phase), CCNE-CDK2 (G1/S transition), CCNA-CDK2/1 (S and G2 phase and S/G2 transition) and CCNB-CDK1 (G2/M transition and M phase), the activity of which is inhibited by CDK inhibitors[35,48,49]. The CCNB-CDK1 complex is not expressed in cell cycle-arrested adult CMs, and it is also not required for CM hypertrophy[49]. In our study, we found a decrease in the expression of CCNB-CDK1 in iPSC-CMs from the MM + NP + ES group. Despite this decrease, these cells exhibited ongoing DNA synthesis, an increased proportion of polyploid CMs and higher cell volume and granularity. The downregulated CCNB-CDK gene expression is associated with the upregulation of CDK inhibitor p21 and MEIS1 that are negatively regulated by TBX20[50]. Better understanding of how MAPK/PI3K-AKT signalling controls cell cycle progression, including the expression of TBX20, MEIS1 and p21, may provide valuable mechanistic insights to promote iPSC-CM maturation or to stimulate adult CM regeneration.

Another important finding of our study is that NP and ES synergistically induce the maturation of different ion channels. We found that NP combined with MM strongly increased $I_{Kr}$ and $I_{K1}$, but had little effect on $I_{to}$, $I_{Na}$ and $I_{Ca-L}$. However, the addition of ES to MM + NP led to significant changes in all currents (larger $I_{Na}$, $I_{to}$, $I_{K1}$, and $I_{Kr}$, but smaller $I_{Ca-L}$) and maturation of electrophysiology (more negative RMP, shorter APD, higher Vmax, APA, CV and spike amplitude, and the 'notch-and-dome' AP morphology) in iPSC-CMs, similar to adult CMs[9,33,51]. The $EC_{50}$ values for isoprenaline chronotropy in this study fell within the nanomolar range (1-47 nM) of $EC_{50}$ values reported for

EHTs cultured in MM[16,22]. Slightly higher $EC_{50}$ values have been reported for EHTs subjected to pacing (98 nM)[28], and for foetal cardiac tissue (30 nM)[28] and adult human heart slices (180 nM)[52]. Notably, significant differences in $EC_{50}$ values for isoprenaline chronotropy were observed in two different iPSC lines (1 nM vs. 47 nM)[16]. The enhanced electrophysiological functionality of iPSC-CMs contributes to their improved predictive value for risk assessment of cardioactive drugs, especially in the case of multi-channel blockers such as verapamil, ranolazine or alfuzosin[15,16]. Several studies reported that verapamil inhibits the beating activity of iPSC-CMs[15,16,53], probably because depolarisation in immature iPSC-CMs does not rely exclusively on $I_{Na}$, as in adult CMs, but also on $I_{Ca-L}$[15]. In line with these studies[15,16], our data demonstrate that increased maturation reduces verapamil-induced beating arrest of iPSC-CMs. The RNA-seq data suggest that the increased $I_{to}$, $I_{K1}$, and $I_{Kr}$ and the decreased $I_{Ca-L}$ in the MM + NP + ES group may be due to the upregulation of genes encoding different potassium channels and the downregulation of genes encoding LTCCs. In addition, the downregulation of PI3K-AKT signalling may contribute to the reduced $I_{Ca-L}$ densities[54]. Furthermore, the similar $Ca^{2+}$ transient amplitudes of all groups together with the decreased $I_{Ca-L}$ in iPSC-CMs from MM, MM + NP and MM + NP + ES groups in comparison to the B27 condition suggest an enhanced excitation-contraction coupling gain, an important indicator of improved calcium handling[23]. The colocalisation of RYR2 with α-actinin and the ability to adapt to high-frequency stimulation support an improved calcium handling, especially in CMs from the MM + NP + ES group. Future studies should investigate how the automaticity of iPSC-CMs in the MM + NP + ES is affected, which is controlled by the coupled system of $Ca^{2+}$ and membrane clocks[55] and may involve T-type calcium channels and HCN channels[56].

Interestingly, we observed no changes in the expression of $SCN5A$, coding for the pore-forming α-subunit of the sodium channel ($Na_V1.5$), but an upregulation of $SCN1B$ and a downregulation of $SCN3B$, which encode the β-subunits ($Na_V$-β1/3) of the sodium channel that interact with the α-subunits[57]. While $SCN1B$ is highly expressed in adult CMs, $SCN3B$ is highly expressed in the embryonic heart[58]. Previous studies have shown that co-expression of $SCN1B$ with $SCN5A$ increases the density of $I_{Na}$[58], and the β1-subunit modulates the cell surface localisation, gating, and kinetics of α-subunits[59]. In addition, $Na_V$-β1 also regulates voltage-gated potassium channels, including $K_V4.3$ and associates with the cardiac intercalated disc proteins N-cadherin and Cx43[59], contributing to the enhanced electrical signal conduction. Future studies should focus on whether/how the downregulation of MAPK/PI3K-AKT signalling in iPSC-CMs regulates the gene expression related to ion channel (for example, $K^+$ and $Na^+$ channels) maturation and function[60] as well as the formation of Cx43 gap junction plaques[61] and intercalated discs[43,62].

Finally, when we compared the expression pattern of marker genes, important for ventricular CM maturity, cardiac ion channels, and genes related to cell cycle activity and mitochondrial development in iPSC-CMs of the MM + NP + ES group with published RNA-seq

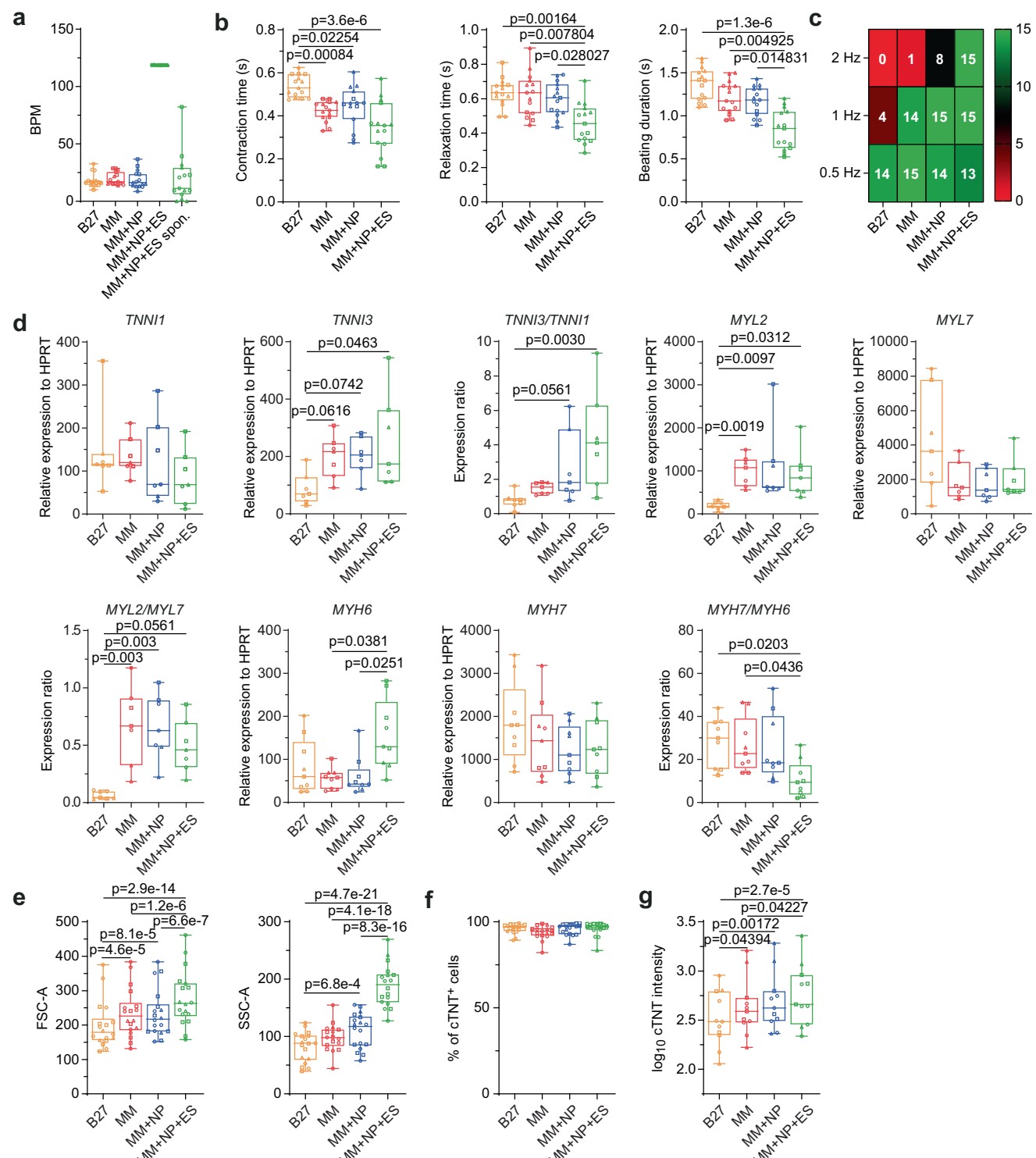

datasets of human foetal ventricle and adult heart samples, we found that the maturity level of iPSC-CMs was intermediate between that of human foetal and adult CMs (Supplementary Fig. 6).

Taken together, we demonstrate that the combined application of MM, NP and ES synergistically induces structural, electrophysiological, metabolic and functional maturation of iPSC-CMs and provide first insights into the mechanism driving advanced maturation of iPSC-CMs. Cultivation in FA-enriched MM strongly improves mitochondrial development and electrophysiological functionality of iPSC-CMs. Although the addition of NP to MM had little effect on the gene expression profile, it induced a specific increase in $I_{K1}$ and $I_{Kr}$ current

densities and cell alignment. The MM + NP + ES combination induces molecular changes that occur during cardiac development, leading to increased structural maturation, polyploidy, improved mitochondrial development, and current patterns of $I_{Na}$, $I_{to}$, $I_{K1}$, $I_{Kr}$ and $I_{Ca-L}$ more similar to adult human CMs. These changes translated into an altered sensitivity of iPSC-CMs to cardioactive drugs, suggesting the efficacy of our maturation approach to improve the predictive power of iPSC-CMs in drug screening. Furthermore, the improved maturation of iPSC-CMs highlights the potential of our combined approach to recapitulate clinical phenotypes that require an advanced development state of iPSC-CMs.

**Fig. 5 | Analysis of contractility and structural development. a** Analysis of beating rate. CMs of the MM + NP + ES group contracted with a beating rate of 120 BPM (beats per minute) during the presence of ES but regained spontaneous beating after ES was discontinued. $n = 15$ cultures/group of 5 independent differentiations of 3 iPSC lines. **b** Statistical analysis of beating properties: contraction time, relaxation time and beating duration of iPSC-CMs under 0.5 Hz field stimulation. $n = 15$ cultures/group of 5 independent differentiations of 3 iPSC lines. A schematic diagram of two beat traces showing how the parameters were analysed is shown in Supplementary Fig. 2a. **c** Heatmap of the ability of iPSC-CMs to adapt to increasing pacing frequencies from $n = 15$ cultures/group of 5 independent differentiations of 3 iPSC lines. The colour scale represents the number of cultures. Representative beating traces are shown in Supplementary Fig. 2b,c. **d** Expression of denoted marker genes for structural maturation. $n = 7$ (*TNNI1, TNNI3, MYL2, MYL7*) and 9 (*MYH6, MYH7*) independent differentiations of 3 iPSC lines. HPRT is used as a housekeeping gene. **e** Quantification of cell volume (FSC-A) and granularity (SSC-A) in the cTNT-positive CM populations using flow cytometry analysis. $n = 18$ independent differentiations/group of 3 iPSC lines. **f, g** Proportion of cTNT-positive cells (**f**) and quantification of mean fluorescence intensity of cTNT (**g**). $n = 17$ (**f**) and 11 (**g**) independent differentiations/group of 3 iPSC lines. Log transformation was performed for statistical analysis (**g**). The gating strategy used to analyse the flow cytometry data is shown in Supplementary Fig. 3. Symbols denote iPSC lines: circles for isWT7, squares for iWTD2, and triangles for iBM76. Source data are provided as a Source Data file. Statistical analysis was performed using Kruskal-Wallis test with Dunn's multiple comparison test (**b, d**) and linear mixed model (two-sided) with Tukey's correction for multiple comparisons between the 4 groups (**e–f**). Data are presented in box plots indicating median (middle line), 25th, 75th percentile (box) and min and max data points (whiskers).

## Methods

### Directed differentiation and pro-maturation culture of iPSC-CMs

In this study, three human iPSC lines were used, which were reprogrammed from somatic cells of three healthy individuals previously. iWTD2.1 (also known as UMGi001-A.1, FB2-iPS1) and iBM76.3 (UMGi005-A.3, MSC3-iPS3) were generated from dermal fibroblasts and mesenchymal stem cells, respectively, using STEMCCA lentivirus[31,63]. isWT7.22 (UMGi020-B clone 22) was generated from dermal fibroblasts using the integration-free CytoTune-iPS 2.0 Sendai Reprogramming Kit[64]. All three cell lines were authenticated by karyotyping and pluripotency assessment, as published previously[31,64]. Regular mycoplasma testing was conducted via PCR analysis using specific primers (for: 5'-ACACCATGGGAGCTGGTAAT-3' and rev: 5'-CTTCWTCGACTTYCAGACCCAAGGCAT-3'), confirming the absence of contamination. The iPSC generation and application in research were approved by the Ethics Committee of the University Medical Centre Göttingen (21/1/11 and 10/9/15) and TU Dresden (EK422092019).

All iPSCs were cultured on Geltrex (Thermo Fisher Scientific, A1413301) coated 6-well plates in Essential 8 (E8) medium (Thermo Fisher Scientific, A1517001) with daily medium change. Cells were passaged or differentiated when they were ~85% confluent. To initiate differentiation, cells were cultured in RPMI 1640 medium (Thermo Fisher Scientific, 72400021) with Glutamax and HEPES, 0.5 mg/mL human recombinant albumin, and 0.2 mg/mL L-ascorbic acid 2-phosphate and treated with 4 μM CHIR99021 (Merck Millipore, 361559), an inhibitor of GSK3β. After 48 h, CHIR99021 was removed and the cells were treated with 5 μM IWP2 (Wnt antagonist II, Merck Millipore, 681671) for another two days. The first beating cells were detected on day 8 post differentiation. From day 8, cells were cultivated in B27 medium containing RPMI 1640, supplemented with 1x B27 with insulin (Thermo Fisher Scientific, 17504044).

On day 15 after differentiation, the cells were digested. Cells were first incubated with 1 mg/mL collagenase B (Worthington Biochemical, CLSAFB) for 1 h at 37 °C, then detached iPSC-CM clusters were gently collected in a 15-mL Falcon tube and dissociated with 0.25% trypsin/EDTA (Thermo Fisher Scientific) for 8 min. Dissociated iPSC-CMs were resuspended in cardio-digestion medium (80% B27 medium, 20% foetal calf serum, and 2 μM thiazovivin (Merck Millipore, 420220)). The resuspended cells were seeded into Geltrex-coated 6-well plates for the B27, MM, MM + ES and DMEM + B27 groups, or onto Geltrex-coated ø25 mm nanopatterned (NP) coverslips (NanoSurface Coverglass, Curi Bio) in 6-well plates for the MM + NP and MM + NP + ES groups at a density of 300,000-500,000 cells/well. NP coverslips are made of glass coverslips covered with a structured polyethylene glycol diacrylate polymer (800 nm groove width, 800 nm ridge width, and 600 nm height) with a surface stiffness of approximately 7 mPa[25]. Cells were maintained in B27 medium for 6 days with medium changes every 2 days. On day 21, all the non-B27 groups were switched from B27 medium to maturation medium (MM, Supplementary Table 1) for 7 days, with medium changes every 2 days. On day 28, the MM + NP + ES and MM + ES groups were subjected to 2 Hz electric field stimulation (2 ms pulse duration, 8 V) using a C-Pace EP (IonOptix) together with a 6-well C-dish (IonOptix) for 14 days. On day 42 post differentiation, cells from all four groups were harvested directly for analysis or dissociated, replated and cultured for another 7 days for further analysis (Fig. 1a).

### Patch clamp analysis

iPSC-CMs at day 42 from the four groups were dissociated using a previously described method[65,66]. Briefly, for MM + NP and MM + NP + ES groups, NP coverslips with cells were transferred to a 3.5-cm dish and then treated for 10 min with 2 mL of 20 U/mL papain (Sigma-Aldrich, 76220) dissolved in 1.1 mM EDTA-buffered B27 medium containing 2.5 μM blebbistatin (Sigma-Aldrich, B0560). The cells were gently centrifuged at $50 \times g$ for 1 min. After aspirating the supernatant, the cell pellet was gently resuspended in B27 medium containing 2.5 μM blebbistatin and stored at 4 °C until measured.

All automated patch-clamp experiments were performed at room temperature (RT) using Patchliner Quattro with PatchControlHT software (Nanion technologies GmbH) with low (for $I_{Na}$, $I_{K1}$, and $I_{Ca-L}$) and medium resistance (for $I_{to}$) NPC-16 chips. The intracellular pipette and extracellular bath solutions for $I_{Na}$, $I_{K1}$, $I_{Ca-L}$, and $I_{to}$ are listed in Supplementary Table 4. To record $I_{Na}$, cells were depolarised from a holding potential of −100 mV using voltage steps from −80 to +70 mV for 20 ms in 5 mV steps. The sweep interval was 2 s. Nifedipine (10 μM) was used to block $I_{Ca-L}$. $I_{K1}$ was recorded using test potentials of 2 s duration between −130 and 10 mV from a holding potential of −40 mV. The sweep interval was 10 s. The protocol was repeated in the presence of 0.5 mM $BaCl_2$ and the $Ba^{2+}$-sensitive current was calculated as $I_{K1}$. To record $I_{Ca-L}$, cells were depolarised for 100 ms to voltages between −80 and +60 mV from a holding potential of −90 mV, and the sweep interval was 3 s. $I_{to}$ was recorded by increasing the test potential from −40 mV to +60 mV in 10 mV steps from a holding potential of −90 mV with a 20 ms pre-pulse to −35 mV to inactivate $I_{Na}$. Each pulse lasted for 600 ms, and the sweep interval was 10 s. $CdCl_2$ (0.5 mM) was used to block sodium and calcium currents.

For action potential (AP) recordings and $I_{Kr}$ measurements using manual patch clamp technique, CMs from all four groups were used directly after overnight recovery in cardio-digestion medium in order to minimise the time that CMs from the MM + NP and MM + NP + ES groups spent in non-NP and non-ES conditions. The pipette and extracellular solutions used for AP and $I_{Kr}$ recordings are listed in Supplementary Table 4. All manual patch-clamp experiments were performed at RT using a ruptured whole-cell patch clamp with a HEKA EPC10 amplifier and Patchmaster (HEKA Elektronik).

To assess resting membrane potential (RMP), maximum upstroke velocity (Vmax), and action potential amplitude (APA), spontaneous APs were recorded in Tyrode's solution without current injection. To

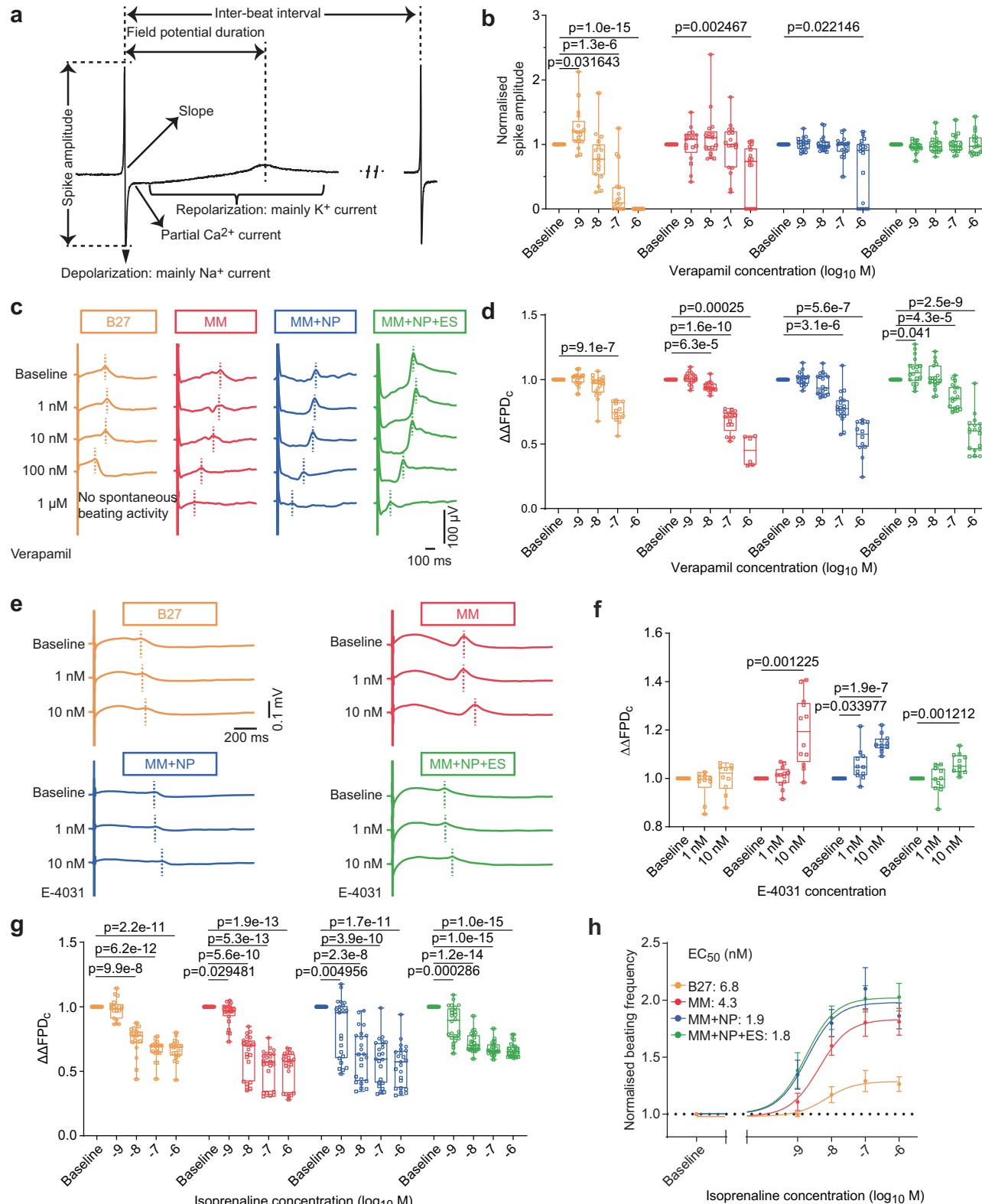

assess AP duration (APD), a negative current was injected into the CMs to maintain the RMP at approximately −80 mV prior to application of 0.5 Hz pacing stimulation. Signals were filtered with 2.9 and 10 kHz Bessel filters. At least 5 consecutive stable spontaneous APs and paced APs were averaged to determine RMP, Vmax, APA and APD at 90% repolarisation ($APD_{90}$) using Fitmaster (HEKA Elektronik) and Lab-Chart 8 software (ADInstruments).

The holding potential of the $I_{Kr}$ recording was set at −50 mV. $I_{Kr}$ was recorded using test potentials of 2.5 s duration from +40 to −40 mV in 10 mV decrements. This was followed by a 4 s phase at −40 mV to elicit the $I_{Kr}$ tail current. The pulse interval for each sweep was 10 s. $I_{Kr}$ was defined as the E-4031-sensitive current by subtracting the current recorded after application of 1 μM E-4031 from the current recorded before application.

**Fig. 6 | Maturation status of iPSC-CMs affects their drug response. a** Schematic diagram of FP traces showing how the data was analysed. **b** Quantitative analysis of the effect of verapamil on spike amplitude. $n = 17$ (B27, MM) and 18 (MM + NP, MM + NP + ES) cultures derived from 3 independent experiments using 2 iPSC lines. Experimental design is shown in Supplementary Fig. 4. **c** Representative averaged field potential (FP) traces of verapamil-treated iPSC-CMs showing the FPDc (FP duration corrected by Fridericia's formula) shortening. **d** Quantitative analysis of the effect of verapamil on FPDc shortening (ΔΔFPDc). $n = 17$ (B27, MM) and 18 (MM + NP, MM + NP + ES) cultures derived from 3 independent experiments using 2 iPSC lines. **e** Representative traces illustrating the FPDc prolongation induced by increasing concentrations of E-4031. **f** Quantitative analysis of the effect of E-4031

on FPDc. $n = 10$ (B27), 12 (MM), and 11 (MM + NP, MM + NP + ES) cultures from 2 independent experiments. **g**, **h** Quantitative analysis of concentration-dependent effect of isoprenaline on FPDc (**g**) and beating rate (**h**). $n = 18$ (B27), 23 (MM, MM + NP), and 24 (MM + NP + ES) cultures derived from 3 (B27) or 4 (MM, MM + NP, MM + NP + ES) independent experiments of 2 iPSC lines. Data are normalised to the respective baseline of each group (**b**, **d**, **f**, **g**, **h**). Symbols in (**b**, **d**, **f**, **g**) denote iPSC lines: circles for isWT7, and squares for iWTD2. Source data are provided as a Source Data file. Statistical analysis using two-way ANOVA with Dunnett's post-test. Data are presented as mean ± 95% CI (**h**) and in box plots indicating median (middle line), 25th, 75th percentile (box) and min and max data points (whiskers) in (**b**, **d**, **f**, **g**).

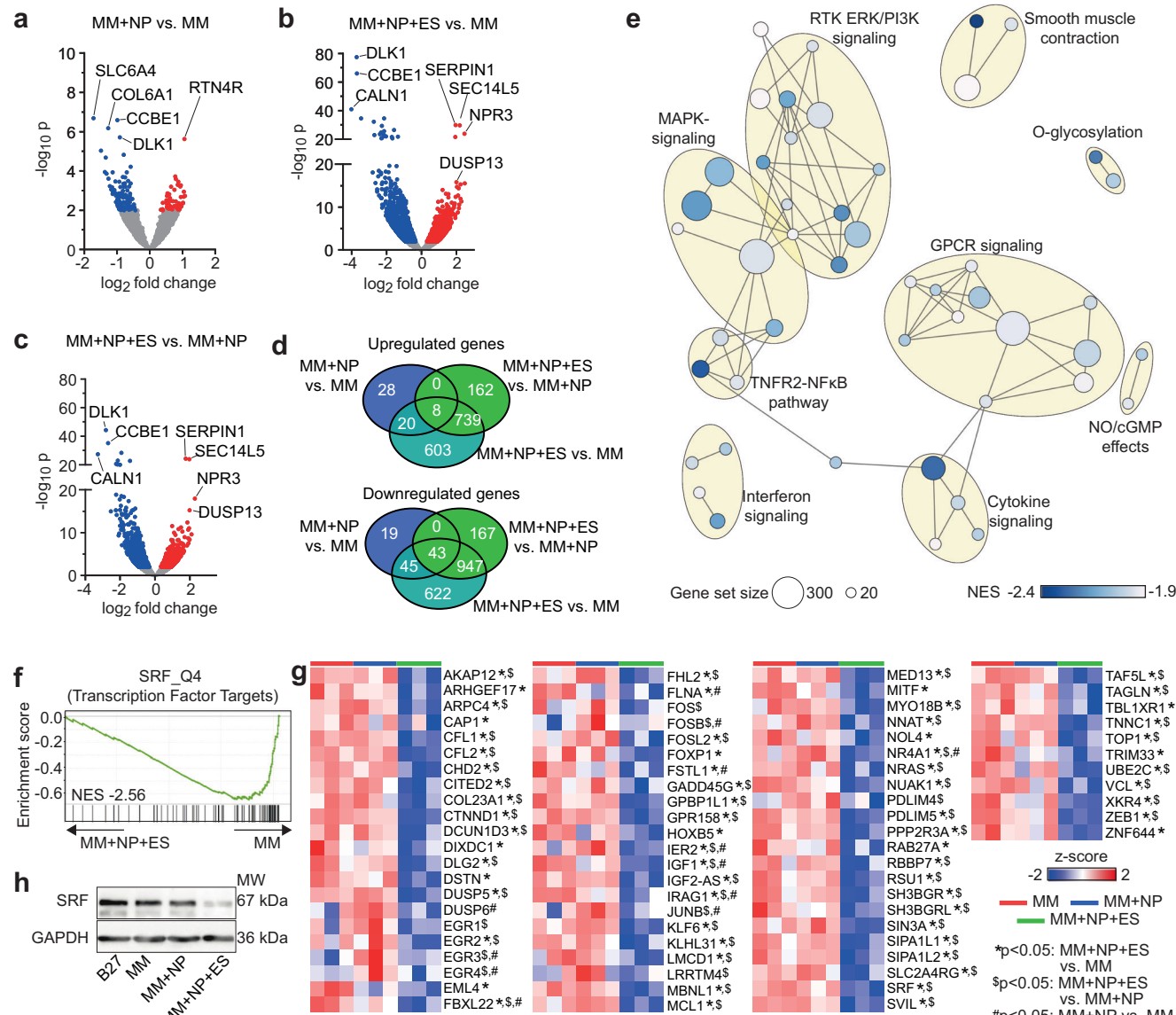

**Fig. 7 | RNA sequencing of iPSC-CMs cultivated under MM, MM + NP and MM + NP + ES conditions. a**–**d** Volcano plots (**a**–**c**) and Venn analyses (**d**) of significantly differentially expressed genes (DEGs, $p < 0.01$) between iPSC-CMs from MM, MM + NP and MM + NP + ES groups. **e** Enrichment map illustrating clustered pathways identified in gene set enrichment analysis (GSEA) based on canonical pathway database. The colour scale represents normalised enrichment score (NES). Pathways were filtered based on max. size of 500 genes, NES ≤ −1.9, false discovery

rate (FDR) $q$-value ≤ 0.2 and cluster size of ≥ 2 pathways. **f** Enrichment plot of SRF_Q4 gene sets obtained with GSEA. **g** Heatmaps of the expression of SRF target genes significantly downregulated in MM + NP + ES. The colour scale represents z-score. **h** Western blot of total SRF. $n = 4$ independent experiments using 2 iPSC lines. Source data are provided as a Source Data file. Statistical analysis was performed using the Wald test of DESeq2 (two-sided) in (**a**–**c**, **g**). Exact $p$ values in **g** are provided in the Source Data file.

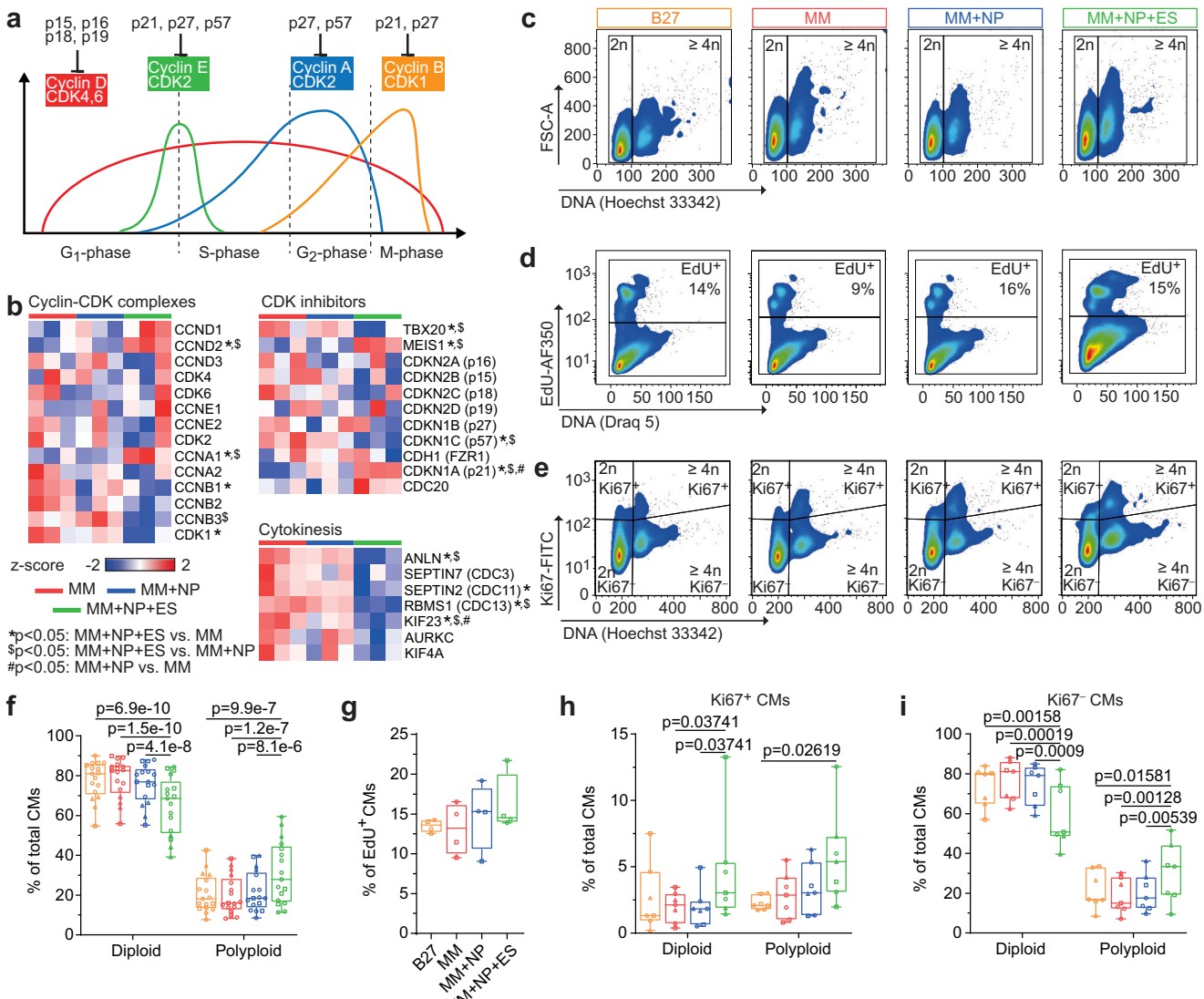

**Fig. 8 | Cell cycle regulation of iPSC-CMs cultivated under MM, MM + NP and MM + NP + ES conditions. a** Scheme illustrating the expression of cyclin-CDK complexes and respective inhibitors during cell cycle. **b** Heatmaps of cyclin-CDK complexes, CDK inhibitors, and cytokinesis related genes including those significantly differentially expressed genes among the three groups. The colour scale represents z-score. **c**–**e** Flow cytometry plots showing cellular DNA content (**c**), DNA synthesis activity (**d**), and Ki67 activity (**e**). **f** Quantification of diploid and polyploid iPSC-CMs (cTNT-positive populations). Data from 17 independent experiments of 3 iPSC lines. **g** Proportion of EdU-incorporated iPSC-CMs. Data from 4 independent experiments of 2 iPSC lines. **h**, **i** Quantification of diploid and

polyploid CMs in Ki67-positive (**h**) and negative (**i**) populations. Data from 7 independent experiments of 3 iPSC lines. Symbols in (**f**–**i**) denote iPSC lines: circles for isWT7, squares for iWTD2, and triangles for iBM76. Source data are provided as a Source Data file. Statistical analysis was performed using the Wald test of DESeq2 (two-sided) in (**b**) or using linear mixed model (two-sided) with Tukey's correction for multiple comparisons between the 4 groups (**f**–**i**). Exact *p* values in (**b**) are provided in the Source Data file. Data are presented in box plots indicating median (middle line), 25th, 75th percentile (box) and min and max data points (whiskers) in (**f**–**i**).

## Multi-electrode array

All multi-electrode array (MEA) recordings were performed using a Maestro Edge equipped with AxIS Navigator software (Axion Bio-Systems) at 37 °C, 5% $CO_2$ at a sampling rate of 12,500 Hz. Approximately 200,000 cells were resuspended in 20 µL of cardio-digestion medium and seeded onto the electrode distribution area of Geltrex-coated CytoView 6-well MEA plates (Axion BioSystems). To evaluate the drug response, iPSC-CMs were seeded into Geltrex-coated CytoView 24-well MEA plates (Axion BioSystems) at a density of 25,000 cells/well. Around one hour after seeding, 1 mL of cardio-digestion medium was gently added into every well. iPSC-CMs were recovered for 7 days in the same medium used during the maturation period. To avoid the influence of different media during recording, iPSC-CMs of all conditions were incubated in MM for one hour before

starting measurements. MEA drug testing was performed using a sequential addition protocol with concentration increments after baseline activity recording in each well (Supplementary Fig. 4). Vehicle controls for all conditions were performed on each assay plate. After drug addition, cells were incubated at 37 °C, 5% $CO_2$ for 12 min, and the response was recorded for 2 min. The main metrics including conduction velocity (CV), spike amplitude, spike slope, inter-beat interval, and corrected field potential duration ($FPD_C$, corrected by Fridericia's formula) were further analysed using AxIS Navigator, Cardiac Analysis Tool and AxIS Metric Plotting tool (Axion BioSystems). In addition, we quantified the number of quiescent and arrhythmic cultures after treatment with verapamil, E-4031 and iso-prenaline based on beating rate variation using Cardiac Analysis Tool. Spontaneous beating frequency was defined as the reciprocal

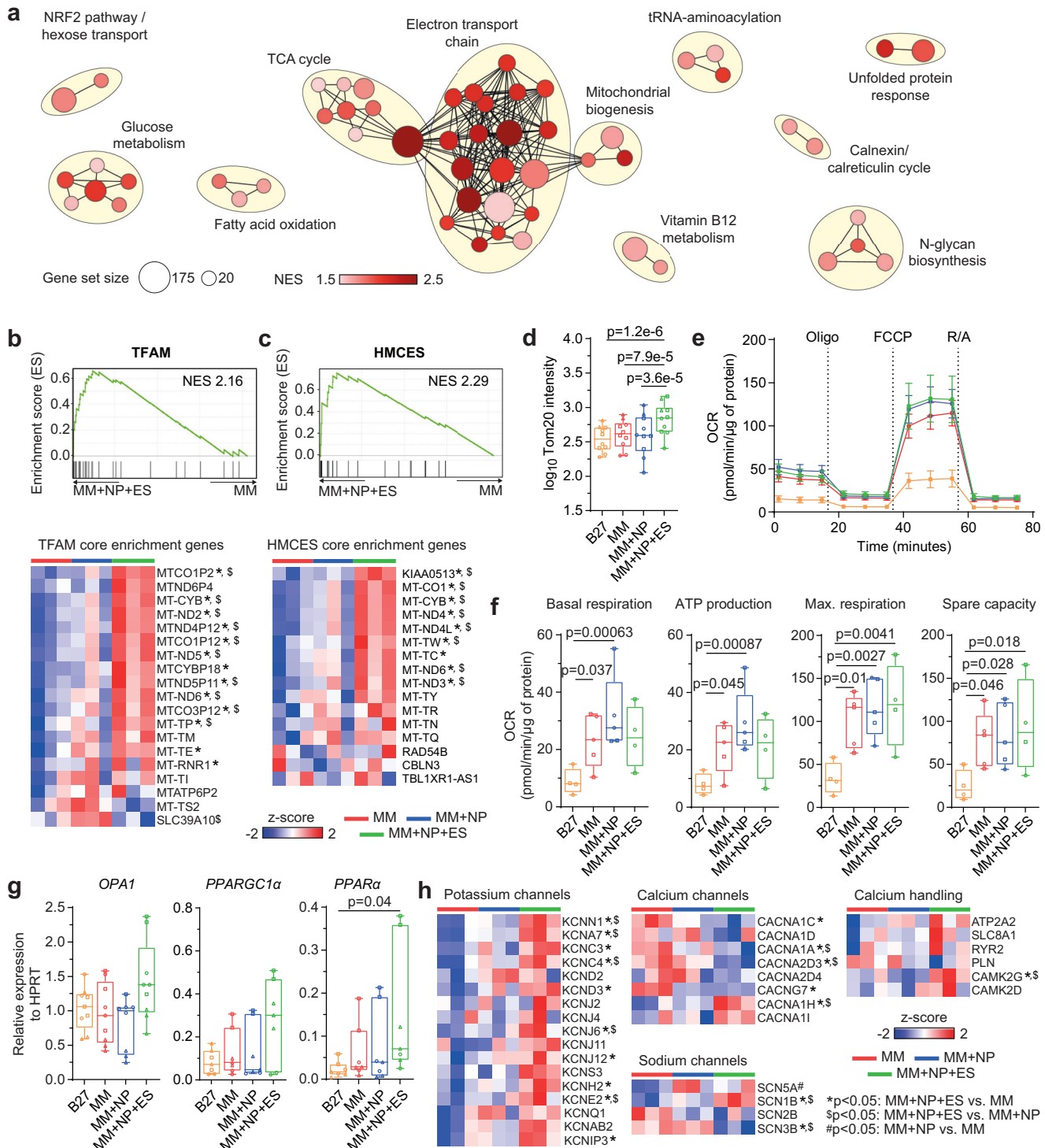

of the averaged inter-beat interval. The mainstream CV values were averaged for CV quantification.

**Calcium transient measurement**

iPSC-CMs at day 42 were dissociated, replated onto Geltrex-coated ø25 mm coverslips at a density of 200,000 cells per well of a 6-well plate and recovered for 7 days in the respective media. For calcium transient measurement using Fura-2[67], cells were loaded with 2.5 µM Fura-2 (Thermo Fisher Scientific, F1221) in B27 medium (B27 group) and MM medium (the other three groups) at 37°C for 30 min and washed twice with the corresponding medium. Cells were incubated for 10 min to achieve complete de-esterification of intracellular Fura-2. For calcium

transient measurement using Fluo-4, cells were loaded with 5 µM Fluo-4 AM (Thermo Fisher Scientific, F14217) and 0.02% (w/v) Pluronic F-127 (Thermo Fisher Scientific, P3000MP) at 37°C for 30 min and washed twice. Intracellular calcium was recorded at 35 °C using a 40x objective on an Olympus IX70 microscope equipped with the IonOptix system with the IonWizard core software (IonOptix). Fura-2 stained samples were excited at 340 and 380 nm with a switching frequency of 200 Hz and the emitted fluorescence was collected at 510 nm. Cytosolic calcium levels were defined as the ratio of fluorescence intensity at 340 and 380 nm (340/380 nm). Fluo-4 stained cells were exited at 488 nm and the emitted fluorescence was collected at 510 nm. $Ca^{2+}$ transients were recorded in Tyrode's solution containing (in mM): NaCl 138, KCl

**Fig. 9 | Upregulation of TFAM and HMCES target genes contributes to mitochondrial development induced by ES. a** Enrichment maps illustrating clustered pathways identified in gene set enrichment analysis (GSEA) based on canonical pathway database. The colour scale represents normalised enrichment score (NES). Pathways were filtered based on max. size of 500 genes, NES ≥ 1.5, false discovery rate (FDR) $q$-value ≤ 0.2 and cluster size of ≥ 2 pathways. **b, c** Enrichment plots and most regulated genes in TFAM and HMCES clusters. The colour scale represents z-score. **d** Quantification of Tom20 intensity detected in cTNT-positive CM populations. $n = 10$ independent experiments with 3 different iPSC lines. Log transformation was performed for statistical analysis. **e, f** Seahorse mean traces (**e**) and determined parameters (**f**) were performed with sequential addition of oligomycin (ATP synthase inhibitor), carbonyl cyanide-p-(trifluoromethoxy) phenylhydrazone (FCCP; mitochondrial uncoupler) and rotenone/antimycin A (complex 1 and 2 inhibitor). $n = 4$ (B27, MM + NP + ES) and 5 (MM, MM + NP) independent experiments using 2 iPSC lines. Orange, red, blue and green lines represent the four

experimental groups: B27, MM, MM + NP, and MM + NP + ES, respectively. **g** Relative expression of *OPA1*, *PPARGC1α* and *PPARα* determined using real-time PCR. HPRT is used as control. Data from 7 (*PPARα*, *PPARGC1α*) or 8 (*OPA1*) independent experiments of 3 iPSC lines. Symbols in (**d, f, g**) denote iPSC lines: circles for isWT7, squares for iWTD2, and triangles for iBM76. **h** Heatmaps of selected genes encoding ion channels including those significantly differentially expressed genes among the three groups. The colour scale represents z-score. Source data are provided as a Source Data file. Statistical analysis was performed using linear mixed model (two-sided) with Tukey's correction for multiple pairwise comparisons between the 4 groups (**d, f**), Kruskal-Wallis test with Dunn's multiple comparison test (**g**), or the Wald test of DESeq2 (two-sided) in (**b, c, h**). Exact $p$ values in (**b, c, h**) are provided in the Source Data file. Data are presented as mean ± SEM (**e**) and in box plots indicating median (middle line), 25th, 75th percentile (box) and min and max data points (whiskers) in (**d, f, g**).

---

4, CaCl$_2$ 1.8, MgCl$_2$ 1, NaH$_2$PO$_4$ 0.33, HEPES 10, and glucose 10 (pH adjusted to 7.3 with NaOH). To normalise Ca$^{2+}$ transient frequency, iPSC-CMs were field stimulated (6 V, 10 ms) at a pacing rate of 0.5 Hz using a MyoPacer (IonOptix). To assess the calcium content, 10 mM caffeine was applied to the CMs under 0.5 Hz pacing. Monotonic transient analysis was performed using LabChart 8 software. For Fura-2 stained cells, diastolic and systolic Ca$^{2+}$ levels, Ca$^{2+}$ transient amplitude (systolic level minus diastolic level), and decay rate (tau) of Ca$^{2+}$ transients were determined. For Fluo-4 stained cells, fluorescence intensities at baseline (F$_0$) and transient peak (F$_1$) were measured and changes in calcium levels (ΔF) were calculated (F$_1$ minus F$_0$). Normalised Ca$^{2+}$ transient amplitude (ΔF/F$_0$), and tau were determined.

## Movie-based contraction analysis

Movie-based contraction analysis was performed on day 42 as previously described[5]. Briefly, movies (1024 × 1024 pixel, 60 FPS, length 30 s, 2 movies per culture, 3 cultures per batch) were recorded using a Hamamatsu Orca Flash 4.0 V3 camera (Hamamatsu) with 60 FPS, 1024 × 1024 pixels resolution. Movies were exported as MPEG4 files and analysed using Maia motion analysis software (QuoData–Quality & Statistics GmbH) using a block size of 10.7 μm (16 pixels), frameshift of 67 ms, and maximum distance shift of 4.69 μm (7 pixels). Contraction and relaxation peaks in the raw beating traces were assessed manually.

## Western blot

On day 42, iPSC-CMs were scraped off, pelleted, snap-frozen in liquid nitrogen and stored at −80 °C. Cells were lysed by homogenisation in RIPA buffer (150 mM NaCl, 50 mM Tris, 1.0% NP-40, 0.5% sodium deoxycholate, 0.1% SDS, 1 mM EDTA, 10 mM NaF, and 1 mM PMSF) supplemented with inhibitors of proteases (cOmplete mini, EDTA-free, Roche) and phosphatases (PhosSTOP, Roche) and incubated for 30 min at 4 °C with gentle rotation. Lysates were centrifuged at 14,000 rpm (19,500 × $g$) for 20 min at 4 °C and protein concentration was determined by BCA assay following the manufacturer's instruction. Proteins were subjected to SDS-PAGE and transferred to nitrocellulose membranes. Membranes were blocked with 5% non-fat milk in TBS-T overnight at 4 °C. Afterwards, the membranes were incubated with primary antibodies overnight at 4 °C, followed by incubation with horseradish peroxidase-conjugated goat anti-mouse or anti-rabbit secondary antibodies (Supplementary Table 5) at RT for 1 h. Proteins were visualised by chemiluminescence using the Super Signal West Dura Chemiluminescent Substrate kit in combination with the Fusion FX Spectra Imaging System (Peqlab) and images were analysed using FusionCapt Advance software (Vilber).

## Flow cytometry

Cells were singularised using collagenase and trypsin, fixed in 4% paraformaldehyde (PFA) for 20 min at RT and stored in PBS containing 1% BSA at 4 °C. For staining, iPSC-CMs were permeabilised in PBS

containing 1% BSA and 0.1% Triton-X for 10 min at RT. Staining was performed with specific antibodies (Supplementary Table 5). cTNT was detected using either mouse anti-cTNT (Thermo Fisher Scientific, MS-295-P1) or directly coupled cTNT-APC (Miltenyi Biotec, 130-120-543). For cTnT and Tom20 staining, cTNT-APC and anti-Tom20 (Santa Cruz, sc-17764) antibodies were used. Negative controls were performed using either the respective secondary antibodies (for samples detected with the non-coupled primary antibodies) or isotype controls (for samples detected with the directly coupled primary antibodies). After incubation with primary antibodies, cells were washed with PBS containing 1% BSA, followed by incubation with secondary antibodies, and Hoechst 33342 (5 μg/mL). To assess EdU-incorporation, PFA-fixed iPSC-CMs were incubated with mouse anti-cTNT antibody in Click-iT™ permeabilisation and wash reagent (Thermo Fischer Scientific, C10645) overnight, EdU click reaction was performed according to manufacturer's instructions, and DNA was stained with Draq5 (abcam, ab108410, 10 μM). To determine the activity of Ki67, iPSC-CMs were stained with antibodies cTNT-APC and Ki67-FITC (Miltenyi Biotec, 130-117-691) for 1 h at 4 °C. DNA was stained with Hoechst 33342. Afterwards, cells were resuspended in PBS containing 1% BSA and analysed on an LSRII or FACS Canto II flow cytometer using FACSDiva software version 8.0.2 (BD Biosciences). At least 10,000 events were recorded for each sample. Flow cytometry data were then analysed using FlowJo v10.10 (BD Biosciences).

## Immunofluorescence staining

For immunofluorescence staining, iPSC-CMs from the B27 and MM groups were dissociated and replated onto Geltrex-coated ø25 mm glass coverslips at day 28 at a density of 200,000 cells per well of a 6-well plate. For staining of RYR2, α-actinin and Hoechst 33342, cells at day 42 were fixed with 4% PFA for 20 min and permeabilised with PBS containing 1% BSA and 0.1% triton-X 100 for 10 min at RT. For detection of Cx43 and Hoechst 33342, iPSC-CMs were fixed in methanol-acetone (7:3, $v/v$, 10 min at −20 °C). After fixation, cells were washed with PBS and incubated in PBS containing 1% BSA at 4 °C for at least 2 h. Incubation with primary antibodies (Supplementary Table 5) was performed overnight at 4 °C in PBS containing 1% BSA. After washing the coverslips in PBS, samples were incubated with secondary antibodies and Hoechst 33342 in PBS containing 1% BSA for 1 h at RT. Coverslips were washed with PBS, deionised water and mounted onto glass slides using Fluoromount-G mounting medium (Thermo Fisher Scientific). Imaging was performed using LSM880 confocal microscope and ZEN software (Carl Zeiss).

To analyse the colocalisation of RYR2 and α-actinin, samples were imaged using an LSM880 confocal microscope (at 60x magnification). Colocalisation was quantified using Cell Profiler v4.2.6 (Broad Institute). For this, composite images were loaded, single channels were separated using the ColorToGray module, colocalisation of the red (RYR2) and green (α-actinin) channels was measured for the entire

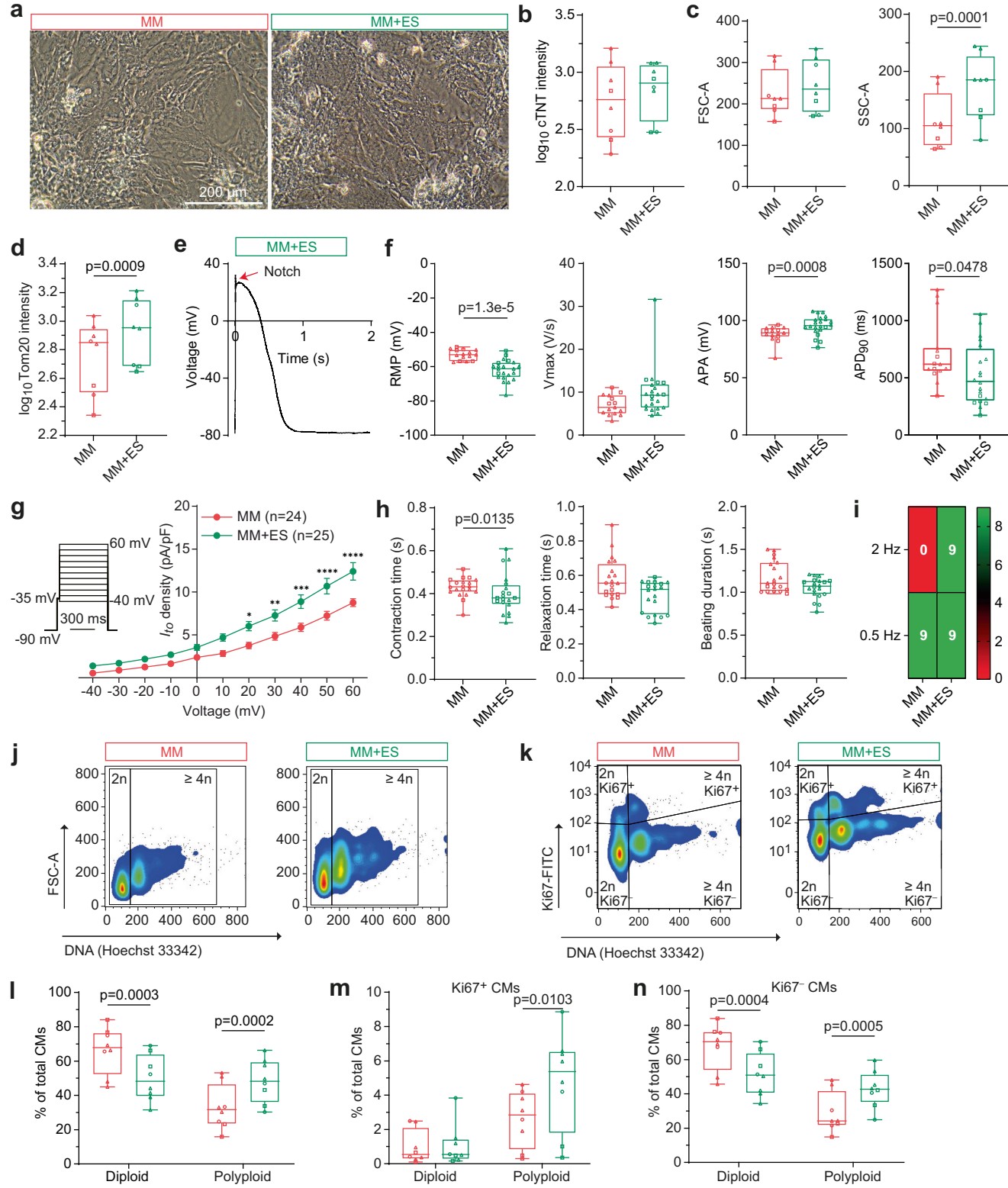

image covering 2-4 cells using the MeasureColocalisation module, and data were exported using the ExportToSpreadsheet function. Colocalisation was measured for 3 independent experiments with 6 images per experiment and conditions.

To study cell alignment, we imaged iPSC-CMs stained for α-actinin and nuclei and the NP surface on the coverslip at different z levels using confocal microscopy. The NP direction was defined as 0° in the MM + NP and MM + NP + ES groups, whereas 0° was randomly defined in the B27 and MM groups. The z-discs and nuclei were identified using

the IdentifyPrimaryObjects module, and z-disk orientation (stained with α-actinin) and the direction of the elongated nuclei (stained with Hoechst 33342) were calculated with respect to 0° using Cell Profiler software.

**Seahorse measurements**

The Seahorse system with Wave Desktop software (Agilent) was used to determine the metabolic activity of iPSC-CMs. Cells were singularised using collagenase B and trypsin on day 42 and replated in 96-

**Fig. 10 | Effect of ES on iPSC-CM maturation. a** Representative morphology of iPSC-CMs under MM and MM + ES conditions. **b**–**d** Quantification of cTNT intensity (**b**), cell volume (FSC-A) and granularity (SSC-A) in the cTNT-positive CMs (**c**), and Tom20 intensity (**d**). $n = 8$ independent experiments with 3 different iPSC lines. Log transformation (**b**, **d**) was performed for statistical analysis. **e** A notch event (red arrow) in an iPSC-CM of the MM + ES group. **f** Quantification of action potential (AP) metrics: resting membrane potential (RMP), maximum upstroke velocity (Vmax), AP amplitude (APA), and AP duration at 90% repolarization (APD$_{90}$). For RMP, Vmax and APA: $n = 15$ (MM) and 22 (MM + ES) cells from 3 (MM) and 4 (MM + ES) independent differentiations of 2 iPSC lines. For APD$_{90}$: $n = 15$ (MM) and 20 (MM + ES) cells from 3 (MM) and 4 (MM + ES) independent differentiations of 2 iPSC lines. **g** Statistical analysis of $I_{to}$ in single cells derived from 5 independent differentiations of 2 iPSC lines. The stimulation protocol is shown as an inset. **h** Statistical analysis of beating properties: contraction time, relaxation time and beating duration of iPSC-

CMs under 0.5 Hz field stimulation ($n = 20$ cultures/group from 5 independent differentiations of 3 iPSC lines). **i** Heatmap of the ability of iPSC-CMs to adapt to increasing pacing frequencies ($n = 9$ cultures/group from 3 independent differentiations of 3 iPSC lines). The colour scale represents the number of cultures. **j**, **k** Shown are flow cytometry plots for DNA content (**j**) and Ki67 activity (**k**). **l**–**n** Quantification of diploid and polyploid cells in total iPSC-CMs (**l**) and in Ki67-positive (**m**) and negative (**n**) populations. $n = 8$ independent experiments with 3 different iPSC lines. Symbols in (**b**–**d**, **f**, **h**, **l**–**n**) denote iPSC lines: circles for isWT7, squares for iWTD2, and triangles for iBM76. Source data are provided as a Source Data file. Statistical analysis was performed using linear mixed model and t-test (two-sided) (**b**–**d**, **l**–**n**), Two-way ANOVA with Sidak's multiple comparison test (**g**), or two-sided Kolmogorov-Smirnov test (**f**, **h**). Data are presented as mean ± SEM (**g**) and in box plots indicating median (middle line), 25th, 75th percentile (box) and min and max data points (whiskers) in (**b**–**d**, **f**, **h**, **l**–**n**).

well Seahorse assay plates (Agilent, 103792-100) at a density of 15,000 cells/well in cardio-digestion medium. The next day, the medium was changed to the appropriate culture medium (B27 medium for the B27 group, MM for three other groups) and cells were recovered for 7 days with medium changes every other day. Seahorse recordings were performed using Seahorse Agilent XF Analyser (Agilent) according to the manufacturer's protocols. The Seahorse XF Cell Mito Stress Test Kit (Agilent, 103010-100) with sequential addition of oligomycin, FCCP and rotenone/antimycin A was used to study mitochondrial respiration. All data were normalised to the total amount of protein per well after lysis of cells in RIPA buffer.

### Real-time PCR
On day 42, the cells ($1 \times 10^6$ cells/well) were washed with ice-cold PBS and scraped from the culture plate or NP coverslips. Two wells of each condition were pooled as one sample that was centrifuged at $2000 \times g$ at 4 °C. All steps were performed on ice. Cell pellets were lysed in 1 mL TRIzol™ (Thermo Fisher Scientific) and homogenised through continuous pipetting (30x) using a metal syringe. Lysates were centrifuged for 5 min at $12,000 \times g$ at 4 °C, the clear supernatant was transferred into a new tube, and 0.2 mL chloroform was added. Samples were mixed thoroughly, incubated for 3 min on ice and centrifuged for 15 min at 4 °C and $15,000 \times g$ to separate phases. The upper phase containing the isolated RNA was carefully collected and mixed with doubled volume of 95% ethanol. The solution was loaded on RNA columns of RNeasy isolation kit (Qiagen), RNA was purified according to manufacturer's protocol, and eluted in RNAse-free water. Concentration was determined using a Nanodrop™ spectrophotometer (Thermo Fisher Scientific). Synthesis of cDNA was performed using the iScript cDNA Synthesis Kit (Bio-Rad) with 200 ng total RNA per reaction (20 μL) according to the manufacturer's protocol. For the detection of gene expression, samples and specific primers (Supplementary Table 6) were prepared using SsoAdvanced Universal SYBR Green Supermix (BioRad) and real-time PCR was performed in Hard-Shell Optical 96-well plates using the CFX96 Real-time PCR system (Bio-Rad). Initial denaturation was done at 95 °C for 30 s, followed by 45 amplification cycles (15 s at 95 °C, 1 min elongation at 60 °C). A melting curve was obtained over the temperature range 65-95 °C. HPRT was used as a reference gene. Efficacies of all primers were determined before use through serial dilution of cDNA and the specificity based on gel electrophoresis as well as melting curves. Primer efficacies were calculated using CFX Manager software (Bio-Rad). For quantification of gene expression, we used Eq. (1) [68] to calculate relative expression of each gene to HPRT.

$$Relative\ expression = \frac{1/(1+efficacy)^{Ct(target)}}{1/(1+efficacy)^{Ct(reference)}} \tag{1}$$

### RNA sequencing and DEG analysis
iPSC-CMs from three independent differentiations of 2 iPSC lines (iWTD2.1, isWT7.22) were used for RNA sequencing analysis. mRNA was isolated from an average of 600 ng total RNA by poly-dT enrichment using the NEBNext Poly(A) mRNA Magnetic Isolation Module (NEB) according to the manufacturer's instructions. Samples were then directly subjected to the strand-specific RNA-seq library preparation workflow (Ultra II Directional RNA Library Prep, NEB). Ligation was performed using the NEB Next Adaptor from the NEB Next Multiplex Oligos for Illumina Kit. After ligation, the adaptors were depleted by XP bead purification (Beckman Coulter), where the bead solution was added to the samples in a ratio of 0.9:1. Unique dual indexing was done during the subsequent PCR enrichment (12 cycles) using amplification primers carrying the same sequence for i7 and i5 index (Primer 1: AAT GAT ACG GCG ACC ACC GAG ATC TAC AC NNNNNNNN ACA TCT TTC CCT ACA CGA CGC TCT TCC GAT CT, Primer 2: CAA GCA GAA GAC GGC ATA CGA GAT NNNNNNNN GTG ACT GGA GTT CAG ACG TGT GCT CTT CCG ATC T). After two further XP bead purifications (0.9:1), the libraries were quantified using the Fragment Analyser (Agilent). Libraries were sequenced on an Illumina NovaSeq 6000 in 100 bp paired-end mode with an average of 50 million fragments per library.

FastQC (http://www.bioinformatics.babraham.ac.uk/) was used to perform a basic quality control of the resulting sequencing data. Fragments were aligned to the human reference genome hg38 with the support of the Ensembl 104 splice sites using the aligner STAR (2.7.10b). Counts per gene and sample were obtained based on the overlap of the uniquely mapped fragments with the same Ensembl annotation using featureCounts (v2.0.1). Normalisation of raw fragments based on library size and testing for differential expression between the different conditions was performed using the DESeq R package (v1.38.3). Sample to sample Euclidean distance, Pearson' and Spearman correlation coefficient (r) and PCA based on the top 500 genes with the highest variance were computed to explore the correlation between biological replicates and different libraries. To identify differentially expressed genes (DEGs), counts were fitted to the negative binomial distribution and genes were tested between conditions using the Wald test of DESeq2 (two-sided). TPM values were generated using Kallisto (v0.46.1).

Gene set enrichment analysis (GSEA) was performed based on normalised count data using GSEA software v.4.3.2[69] and canonical pathways (CP) collection (C2, CP: c2.cp.v2023.2.Hs, https://www.gsea-msigdb.org/gsea/msigdb/human/genesets.jsp?collection=CP) or TFT collection (C3, TFT: c3.tft.v2023.2, https://www.gsea-msigdb.org/gsea/msigdb/human/genesets.jsp?collection=TFT), two Human Molecular Signatures Database (MSigDB) collections provided by the Broad institute with following parameters: weighted scoring, meandiv normalisation, max_probe mode, maximum gene set size 500 genes, minimum set size 15 genes, and 1000 permutations. GSEA results were

applied for the creation of enrichment maps using Cytoscape software according to[70]. Briefly, nodes were included based on false discovery rate (FDR) q-value < 0.2 and normalised enrichment score (NES) ≥ 1.5 for upregulated and ≤ −1.9 for downregulated genes. Edges cut-off was set to 0.375 and pathway cluster names were determined after manual examination of all individual nodes in clusters.

To benchmark the gene expression pattern of iPSC-CMs in the MM + NP + ES group with human heart tissue, our own data and published bulk RNA-seq datasets (GSE62913) from foetal ventricles (GSM1536186, GSM1536187) and adult heart samples (GSM1536192, GSM1536193) have been were aligned using STAR to the homo sapiens reference hg38 and using Ensembl annotation 104. A list of 210 genes, including marker genes for ventricular CMs, ion channels, and genes related to cell cycle activity and mitochondrial development that were affected by ES in our study, was used for comparison. Fragments were counted using FeatureCounts to assign fragments to gene features. The counts table was processed using R and the libraries DESeq2 (v1.38.3) and edgeR (v3.40.2). DEseq2 standard methods were used to normalise the data. Edge methods were applied to the normalised data to correct for possible effects of different library preparation methods. The normalised and corrected data were then visualised using the R library ggplot2.

### Statistical analysis and reproducibility

Statistical analysis was conducted using R Studio v2024.04.2 (Posit Software, PBC) by applying a linear mixed model (*"lmerTest::lmer"* fitting of the raw values), considering experimental groups and cell lines as fixed variables and independent batches as a random variable. Residuals were evaluated for normal distribution based on plots of residuals vs. fitted values and Shapiro-Wilk test (check_normality function), and for homogeneity with Bartlett's test (check_homogeneity function). The analysis using the linear mixed model followed by pairwise comparisons (using *"emmeans(~group) %>% pairs()"*) represents a two-sided statistical test. In detail, the function *emmeans* ('Estimated marginal means') estimates the mean values per group. These are forwarded to the *pairs* function, which performs the pairwise comparisons of the mean values. If more than 2 groups are analysed, Tukey's post-test is performed to correct for multiple comparisons. In case of 2 groups, significance was determined using a two-sided standard t-test.

If the mixed model could not be applied due to violation of normality or homogeneity, statistical analysis was performed using Kruskal-Wallis test with Dunn's multiple comparison test (comparison of > 2 groups), two-sided Kolmogorov-Smirnov (comparison of 2 groups), or two-way ANOVA with Sidak's post-test for multiple comparisons using GraphPad Prism 10, indicated in figure legends. Results were considered statistically significant at $p < 0.05$.

### Reporting summary

Further information on research design is available in the Nature Portfolio Reporting Summary linked to this article.

## Data availability

All data supporting the findings of this study are available in the article, its Supplementary Information and Datasets, and its Source Data. Source data are provided in this paper. The raw sequencing data are publicly available at the Gene Expression Omnibus (GEO) under accession number GSE290322. RNA-seq datasets of human foetal ventricle (GSM1536186, GSM1536187) and human adult heart (GSM1536192, GSM1536193) were obtained from GEO under accession code GSE62913. Source data are provided with this paper.

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

## Acknowledgements
We thank Susann Hoefner from the Flow Cytometry Core Facility and the Core Facility Cellular Imaging (CFCI) of the TU Dresden for excellent technical support, and Dr. Konrad Grützmann from the Bioinformatics Core Unit at Institute for Medical Informatics and Biometry at TU Dresden for excellent support in conducting the statistical analysis. This work was funded by the Free State of Saxony and the European Union (SAB EFRE projects "PhenoCor" with project number 100387678 to A.E.-A. and K.G. and "CardioEpiX" with project number 100685417 to K.G. as well as ESF Plus project "MultiMOD" with project number 100649621 to K.G.), by the Deutsche Forschungsgemeinschaft (DFG, German Research Foundation) under project number 288034826 –IRTG 2251: "Immunological and Cellular Strategies in Metabolic Disease" for project 10 to K.G., by the German Federal Institute for Risk Assessment with the Grant Agreement Number 60-0102-01#00067 - P639 to M.S., and by the Medical Faculty of the TU Dresden (MeDDrive project) to M.S. In particular, X.L. and A.S. were financially supported by the ESF Plus project "MultiMOD" and A.S. and M.H. were financially supported by the DFG project IRTG 2251.

## Author contributions
W.L., M.S., and K.G. conceived the study, managed the project progress, and coordinated the experiments and analyses; W.L., X.L., A.S., S.A., O.G., M.S.P., M.H., R.-P.S., K.F., J.P., Y.U., G.T., P.M., and M.S. performed the experiments; W.L., X.L., A.S., G.T., M.H., M.L., A.D., M.S., and K.G. contributed to data analyses; visualisation was performed by W.L. and M.S.; and P.M., A.E.-A and K.G. provided resources. The paper was prepared by W.L., M.S. and K.G., with feedback from all authors. All authors have read and approved the current version of the manuscript.

## Funding

## Competing interests
The authors declare no competing interests.
