## [Transparent Peer Review file · Nature Communications]

Comprehensive promotion of iPSC-CM maturation by integrating metabolic medium with nanopatterning and electrostimulation

Corresponding Author: Professor Kaomei Guan

Version 0:

Reviewer comments:

Reviewer #1

(Remarks to the Author)

The authors investigated combinatorial effects of techniques to mature iPSC-derived cardiomyocytes (iPSC-CMs). Immaturity of iPSC-CMs using standard culture techniques has been a major limitation. Many techniques have been reported over recent years to induce greater levels of maturation – however, the myriad protocols has left a need for systematic testing of combinatorial effects. The current work proposes that using a combination of “maturation media”, electrophysiologic training (by rapid pacing), and a nano-patterned cell culture substrate together result in heightened maturation. The individual techniques used in this study have previously been reported, including several papers reporting maturation media methods combined with tissue approaches (e.g. PMID 32697997, PMID: 28916735, PMID: 34697315) and pace training (e.g. PMID: 29618819, PMID: 37419294, PMID: 35931080). The results here add an overall incremental understanding of contributors to maturation through systematic study and a focus on the electrophysiologic state of the iPSC-CMs with each protocol. In particular, it is noteworthy that the investigators attained a “spike and dome” pattern in at least a proportion (43%) of iPSC-CMs treated with the combinatorial approach. The findings have implications for applications of iPSC-CMs that require a higher level of maturation, including drug testing for electrophysiologic effects, as demonstrated by the authors (Fig 5).

1. The “maturation media” has DMEM as a base rather than RPMI. The difference in calcium concentration is large (1.8 mM vs 0.4 mM) and could explain many of the observed electrophysiologic differences (independent of fuel source in the media). Given the focus of the authors on electrophysiology, this highly important detail should be experimentally tested to make the causal claim of metabolic effects (vs calcium effects) and should be discussed in the paper (particularly for experiments measuring calcium handling).
2. The nano-patterning appears to have an underwhelming influence on maturation in many of the assays (including RNA-seq, where little/no difference was observed). Given this, some of the key findings would warrant testing in a MM+ES combination to determine whether nanopatterning is really adding any significant benefit. As written, the paper appears to conclude that nanopatterning is required for the combinatorial effect, but since MM+ES alone was not tested the importance of nanopatterning remains unclear.
3. Nanopatterning on a rigid substrate as used here still prevents symmetric contractions across the substrate and is a major limitation of this technique compared to 3D tissues or 2D cell/tissue patterning techniques that allow consistent fractional shortening across the cells/tissue.
4. The methods state that 3 iPSC lines have been used. However, most of the legends report only 2 lines having been used. Moreover, some legends state quite a low number of replicates to be obtained from the number of stated lines/differentiation batches (e.g. 4b legend MM+EP+NP condition only included 9 cells total from 3 differentiation batches each for 2 lines – this would imply each differentiation batch only contributed 1-2 cells) – these sparse numbers could result in inadvertent selection bias. While patch clamping experiments are obviously very time consuming, only data including adequate numbers of cells from multiple batches/lines should be used. Please state specifically the lines used for each experiment. Key results would ideally be shown in 3 lines.
5. The number of lines/batches/replicates used for the RNA-seq experiment are not described (at least not in legend or the methods section).
6. Standard deviations rather than SEM should be used to better show the spread of the data.
7. Methods for relative quantification of genes by qRT-PCR are not clear (Fig 4i) – if done by qRT-PCR, primers may have

different efficiencies which will markedly affect ratios if directly compared. Referencing each to the housekeeping gene may be more appropriate unless primer efficiencies are directly measured, or the RNA-seq data may be better suited to quantify the relative expression effects.

Reviewer #2

(Remarks to the Author)

This is a nice contribution addressing the maturation of human iPSC-CMs with a combination of maturation media (MM), nanopatterned interfaces (NP), and electrical pacing (ES). The experiments are well planned, presenting key data demonstrating the maturation of iPSC-CMs induced by the individual conditions, with the combination of MM + NP + ES being the most impressive. They are expected given the number of publications employing the conditions; however, the detailed genomic and electrophysiology analysis is a step forward for the field. Some minor comments are provided to improve the manuscript.

1. More details are needed on the structure of the commercially obtained nanopatterned (NP) coverslips (NanoSurface Coverglass, Curi Bio). What are the essential features of these interfaces? Related, in Fig. 1b it is important to demonstrate cell orientation relative to the NP surface features, i.e. alignment. This should be quantitative, essentially measuring the degree deviation between cells and surface features. What is the difference in the sarcomere structure on NP surfaces, both with and without ES? Essentially, I would like to see data on how the NP surface improves 2D tissue alignment, alters sarcomere structure.

2. "However, it is worth noting that 3D-tissue generation is challenging and the experimental throughput is lower compared to 2D-cultures." Although this statement is true, it ignores the comprehensive data generated showing 2D surfaces are inferior to 3D microtissues. Refs. 16 & 17 demonstrated this, among other papers, including Ronaldson-Bouchard (which was omitted - Ronaldson-Bouchard, K., Ma, S.P., Yeager, K. et al. Advanced maturation of human cardiac tissue grown from pluripotent stem cells. *Nature* 556, 239–243 (2018). <https://doi.org/10.1038/s41586-018-0016-3>). One main reason for this is that the 2D monolayers must be attached to the surface, where there is a significant mechanical mismatch, thus altering contraction behavior. I suggest the authors address the 2D vs 3D issue in the context of improved alignment, in part supported by the recommendations in #1.

3. Present calculated IC50 or EC50 from the drug experiments. How do these data compare to other studies using iPSC-CMs using maturation media and pacing? How do they compare to human heart tissue slices.

4. Were any arrhythmia-like events observed, and if so, were there differences between the treatments.

5. The discussion does not compare their results to previously published work, especially those that use similar maturation media.

Minor

Fig. 1c – The APD90 is very long and not physiological for all the conditions, except MM+NP+ES. This is a departure from many studies in 2D showing APD in a more physiological range, ~ 400-500 ms.

Fig. 3d – pacing inset difficult to understand.

429 – line " These findings align with previously reported data on the positive effects of FA supplementation^{13,20}. Refs 16 and Mills' PNAS 2017 paper should be cited here, probably others as well. (<https://www.pnas.org/doi/full/10.1073/pnas.1707316114>)

Reviewer #3

(Remarks to the Author)

The authors submit a manuscript investigating the role of patterning, metabolic substrate and electrical pacing on maturation. Overall the authors present some interesting findings relating to differences in morphology, electrophysiology, transcriptomes, drug responses and cell cycle. This is interesting for the field whereby researchers are trying to further mature hPSC-CM for accurate modelling. Overall the study seems well executed and written, although, there are a number of concerns that need to be addressed in order to determine whether their findings are impactful. Furthermore, for a journal of *Nature Communications* standing, it would be of substantial importance to validate modelling of an important disease/drug/phenotype that can be uncovered by their more mature system over regular hPSC-CM culture. Without such a demonstration the speculation that these results can be extrapolated into eg patient specific modelling are currently unsubstantiated.

Major

1. Others have seen better responses in 2D hPSC-CM to drugs in comparison to the B27 control group (eg. PMID: 28153730). Therefore, this reviewer wonders whether this group in the paper is especially immature and how significant the maturation in the other groups actually is?
2. For the n numbers were approximately the same n numbers obtained for both B27 controls and the different conditions

within each experiment before pooling of the experiments? If different for each experiment, the results will be subject to potential batch effects which may not be appropriately controlled for.

3. I am not sure the appropriate statistical tests were performed as some data looks skewed and some do not seem to have equal standard deviations. That is of considerable concern as I am not sure some of the comparisons are actually statistically different if the appropriate tests are used.

4. Figure 1G-I - Cells are flat and pancake-like rather than aligned like mature CM. The representative section of a small area of the sarcomeres is misleading. With the variability in Fig 1H quantification it's difficult to determine whether the changes are biologically meaningful. Also the CX43 staining is adjacent rather than end-end in Fig 1I.

5. Reduction of proliferation occurs with maturation condition in particular fatty acids-exercise (PMID: 28916735). This would decrease cell number and increase cell size, rather than a direct impact of eg. hypertrophy. This explanation for the changes is highlighted by the cell cycle analysis in Fig 7, and therefore the likely mechanism of these changes.

6. For the gene expression analyses the genes in the heatmaps seem to be cherry-picked, and given the variability in the expression for many of them it's difficult to determine which are and are not regulated. It would be preferable to only present and highlight regulated genes. Additionally, the gene expression could be benchmarked to human heart tissue so that the reader can understand what the remaining deficits in maturity are.

7. The qPCR results seem to be highly variable. Are the differences also seen in the RNA-seq? What do the data points represent? Are they individual wells, were wells scraped and pooled together? Expanding on the details of sample preparation in the first part of the 'Real-time PCR' methods section would also help (approximate quantity of cells, volume of TRIzol, and chloroform etc)

8. The observation of Cx43 in the perinuclear region is interesting. GJA5 is translated by ribosomes in this region to generate Cx43, which is then trafficked to the cell membrane (PMID: 28576298). Could trafficking of proteins also be another mechanism that is being improved in the MM + NP and MM + NP + ES groups? Does the transcriptomics data provide any insight?

Minor

9. Use of colour on graphs. Appreciate the consistent use of colours throughout the paper to separate the different groups that were tested, it greatly enhances readability and interpretability. However, in some graphs the formatting choices obscure the data. The best example I can see is Fig 5g. The boxplots with error-bars and individual data points without the use of semi-transparency should be avoided.

10. Lines 62-66. Could add mention of abundance, localization, lack or presence of t-tubules, as additional differences between mature cardiomyocytes and iPSC-CMs. Or add reference to reviews where that is discussed.

11. Lines 109-112. Figure 1g-h. The authors' statements that "the α -actinin/RYR2 co-localization was enhanced in both the MM and MM + NP groups..." is not supported by the statistical comparisons in Figure 1h. Only in the MM + NP + ES group was there a statistically significant increase in co-localization.

12. Line 141. The "(43%)" was confusing

13. Line 163-164. Fig 2a. The "representative" trace chosen for the MM+NP+ES group has a RMP of about -75mV (it is difficult to tell due to the spacing of the ticks on the y-axis), but the average RMP for the same group shown in Fig 1b is about -65mV.

14. Line 163. A summary table of in the extended data describing all of the parameters of these experiments will enhance the interpretation of the data and could be a valuable resource for others doing similar work.

15. Fig 4d. The units appear to be the measured Fura-2 ratio from Fig 4c, for clarity can this be added to the y-axes or to the figure legend.

16. Line 282 and Fig 5h. Inconsistent use of the name for isoprenaline/isoproterenol. It is best to pick one and stick to it throughout the manuscript to prevent any confusion for non-experts.

17. Line 318. The authors refer to a "transcription factor target database." It is not obvious what this is to this reviewer, can more details be provided?

18. Line 394. T-type calcium channels are upregulated, can the authors provide comment in the discussion on the potential impact of this on automaticity of the iPSC-CMs in context of the other changes that were seen? Perhaps also include mention of HCN1 and HCN4 which are known to be more abundant in iPSC-CMs than (ventricular) adult cardiomyocytes (PMID: 37028405).

19. Fig 8h. "SCL8A1" should be "SLC8A1"?

Version 1:

Reviewer comments:

Reviewer #1

(Remarks to the Author)

The authors have comprehensively addressed my critiques. They have added new experiments to add a comparison for MM without NP, comparison of increased media calcium concentration alone (via DMEM), and increased replicates. These additions make the manuscript much stronger to support the conclusions. As noted in my initial review, many of the findings have been previously reported, but the investigators here have provided more electrophysiologic measurements, which will be a nice addition to the iPSC-CM maturation field. I do think it would be very worthwhile to work the major effect of calcium and MM-ES (with minor or no effect of NP) in electrophysiologic maturation into both the abstract and introduction since the summaries currently still make it appear as though the NP is important to a combinatorial effect. In particular, calcium should be highlighted given the investigators focus on electrophysiology and the fact that calcium seemed to be the major driver of some of the EP parameters (this has not been emphasized much in the existing literature).

Reviewer #3

(Remarks to the Author)

The authors have addressed all of my comments.

Reviewer #4

(Remarks to the Author)

This manuscript defines a rigorous study to evaluate separate and combined effects of methods for driving cardiomyocyte maturation. I feel the authors have responded appropriately to all reviewer comments and have greatly improved the manuscript, particularly by consulting with a statistician and clear evaluating and explaining the appropriate statistical tests, evaluating alignment and combinatorial effects of patterning, performing additional evaluation of the RNAseq data, and including more specific and complete data in supplementary data.

POINT-BY-POINT RESPONSE TO REVIEWERS' COMMENTS

GENERAL REMARKS

We would like to thank the reviewers for their constructive comments and suggestions on our manuscript. In order to comprehensively address the comments, we have performed new experiments and included new results in the extensively revised manuscript:

- We performed additional differentiations to obtain data from all 3 iPSC lines and to increase the number of replicates. The new data were integrated into new figures, including α -actinin/RYR2 colocalisation; patch clamp measurements (I_{Na} , I_{Ca-L} , I_{to} , I_{K1} , I_{Kr}); contraction analysis (beating duration, contraction time, relaxation time, response to pacing); real-time PCR (*TNNI1*, *TNNI3*, *MYL2*, *MYL7*, *MYH6*, *MYH7*, *OPAI1*, *PPARGC1A*, *PPARA*); and flow cytometry analysis (FSC-A, SCC-A, % of cTNT-positive cells, cTNT intensity, Tom20 intensity, ploidy, Ki67).
- We investigated the effects of high calcium concentration on I_{Na} , I_{Ca-L} and calcium transients.
- We studied the effects of MM+ES on the maturation of iPSC-CMs, including flow cytometry analysis (FSC-A, SCC-A, cTNT intensity, Tom20 intensity, ploidy, Ki67), electrophysiological analysis (action potential and I_{to}), and contractile properties.
- We quantitatively measured the degree of deviation between cells and NP.
- We compared our RNA-seq data with available datasets from human fetal and adult hearts.
- We consulted a statistical expert on campus, performed new statistical analyses throughout the manuscript, and presented the data in a more clear way in the revised figures to better show the spread of the data.

REVIEWER COMMENTS

Reviewer #1 (Remarks to the Author):

The authors investigated combinatorial effects of techniques to mature iPSC-derived cardiomyocytes (iPSC-CMs). Immaturity of iPSC-CMs using standard culture techniques has been a major limitation. Many techniques have been reported over recent years to induce greater levels of maturation – however, the myriad protocols has left a need for systematic testing of combinatorial effects. The current work proposes that using a combination of “maturation media”, electrophysiologic training (by rapid pacing), and a nano-patterned cell culture substrate together result in heightened maturation. The individual techniques used in this study have previously been reported, including several papers reporting maturation media methods combined with tissue approaches (e.g. PMID 32697997, PMID: 28916735, PMID: 34697315) and pace training (e.g. PMID: 29618819, PMID: 37419294, PMID: 35931080). The results here add an overall incremental understanding of contributors to maturation through systematic study and a focus on the electrophysiologic state of the iPSC-CMs with each protocol. In particular, it is noteworthy that the investigators attained a “spike and dome” pattern in at least a proportion (43%) of iPSC-CMs treated with the combinatorial approach. The findings have implications for applications of iPSC-CMs that require a higher level of maturation, including drug testing for electrophysiologic effects, as demonstrated by the authors (Fig 5).

We appreciate the positive comments.

All mentioned publications have been included in the Introduction section (lines 72-73) and discussed together with our data in the Discussion section (e.g. lines 522-545).

1. The “maturation media” has DMEM as a base rather than RPMI. The difference in calcium concentration is large (1.8 mM vs 0.4 mM) and could explain many of the observed electrophysiologic differences (independent of fuel source in the media). Given the focus of the authors on electrophysiology, this highly important detail should be experimentally tested to make the causal claim of metabolic effects (vs calcium effects) and should be discussed in the paper (particularly for experiments measuring calcium handling).

We thank the reviewer for pointing out this important issue. To investigate the contribution of the high calcium concentration in MM to the maturation of iPSC-CMs, we first added 1.4 mM calcium (CaCl_2) to the B27 medium, but precipitation was observed in the medium, which may be due to the higher phosphate concentration in RPMI compared to DMEM (5.6 mM vs. 0.9 mM), leading to $\text{Ca}_3(\text{PO}_4)_2$ formation.

We then cultured iPSC-CMs in DMEM+B27 medium containing 1.8 mM calcium (Supplementary Table 1) compared to (RPMI+)B27 medium containing 0.4 mM calcium. Please see lines 236-252: “Previous studies reported that high calcium concentration induces positive force-frequency behaviour, physiological twitch kinetics and robust β -adrenergic response of EHT by improving Ca^{2+} handling³⁰. To investigate the contribution of the high calcium concentration in MM to the maturation of iPSC-CMs, we cultured iPSC-CMs in DMEM+B27 medium containing 1.8 mM calcium (Supplementary Table 1) compared to (RPMI+)B27 medium containing 0.4 mM calcium (Fig. 4f-j). Analysis of the calcium transient parameters revealed only slightly increased Ca^{2+} transient amplitudes but significantly decreased tau in iPSC-CMs cultured in DMEM+B27 compared to (RPMI+)B27 (Fig. 4g). Furthermore, caffeine-induced SR Ca^{2+} release was significantly increased in the DMEM+B27 group compared to the (RPMI+)B27 group (Fig. 4h). Interestingly, all changes were similar to those in the MM group (Fig. 4d,e). We also found that cultivation in DMEM+B27 led to an increase in I_{Na} density to 90.4 ± 7.2 pA/pF at -20 mV compared to -42.3 ± 4.7 pA/pF in the RPMI+B27 group (Fig. 4i). $I_{\text{Ca-L}}$ density was significantly reduced in the DMEM+B27 group (-6.2 ± 0.8 pA/pF at 10 mV) compared to the B27 group (-8.4 ± 0.5 pA/pF) (Fig. 4j). Both I_{Na} and $I_{\text{Ca-L}}$ in the DMEM+B27 group are comparable to those in the MM group (Fig. 3a and Fig. 4b). Overall, these results demonstrate the strong influence of high Ca^{2+} concentrations on the electrophysiological properties of iPSC-CMs.”

Fig. 4 **f-j** Effects of calcium concentrations on Ca^{2+} transient amplitude and tau (**g**, $n = 38$ cells/group from 2 iPSC lines), and SR Ca^{2+} release induced by 10 mM caffeine (**h**, $n = 17$ and 15 cells for the (RPMI+)B27 and DMEM+B27 groups from 2 iPSC lines, respectively), as well as I_{Na} (**i**, $n = 43$ and 37 cells for the (RPMI+)B27 and DMEM+B27 groups from 6 independent differentiations of 3 iPSC lines, respectively) and $I_{\text{Ca-L}}$ (**j**, $n = 27$ cells/group from 6 independent differentiations of three iPSC lines). Statistical analysis was performed using Two-way ANOVA with Sidak's multiple comparison test (**b**, **i**, **j**), Kruskal-Wallis test with Dunn's multiple comparison test (**d**, **e**) or Kolmogorov-Smirnov test (**g**, **h**): * $p < 0.05$, ** $p < 0.01$, *** $p < 0.001$, **** $p < 0.0001$. Data are presented as mean \pm SEM (**b**, **i**, **j**) and in box plots indicating median (middle line), 25th, 75th percentile (box) and min and max data points (whiskers) in **d**, **e**, **g**, **h**.

2. The nano-patterning appears to have an underwhelming influence on maturation in many of the assays (including RNA-seq, where little/no difference was observed). Given this, some of the key findings would warrant testing in a MM+ES combination to determine whether nanopatterning is really adding any significant benefit. As written, the paper appears to conclude that nanopatterning is required for the combinatorial effect, but since MM+ES alone was not tested the importance of nanopatterning remains unclear.

Following the Reviewer's suggestion, we performed new experiments to test the effect of MM+ES alone on the maturation of iPSC-CMs. In the revised manuscript, we added a sub-chapter in the Results part "MM+ES alone induces electrophysiological maturation and polyploidy of iPSC-CMs" (lines 468-494) and Fig. 10.

"As MM+NP had little synergistic effect on gene expression profile when compared to MM alone, it is interesting to test whether MM+ES alone can induce the maturation of iPSC-CMs similar to MM+NP+ES. To study this, we compared iPSC-CMs cultured under MM and MM+ES conditions. We observed no significant differences in cell morphology, cTNT intensity, and cell size (FSC-A) (Fig. 10a-c). Consistent with the changes observed in the MM+NP+ES group, iPSC-CMs in the MM+ES group showed increased granularity (SSC-A), and Tom20 intensity (Fig. 10c,d). These data indicate that ES alone enhances structural maturation, whereas the addition of NP may contribute to cell alignment, cell size and cTNT intensity. We observed the 'notch-and-dome' AP morphology in iPSC-CMs (32%) of the

MM+ES group (Fig. 10e) comparable to those in the MM+NP+ES group. iPSC-CMs in the MM+ES group had a more negative RMP, increased V_{max} and APA, and shortened APD_{90} compared to the MM group (Fig. 10f), but with a less extent when compared to the MM+NP+ES group (Fig. 2b). Similarly, I_{to} density was increased in the MM+ES group compared to the MM group (Fig. 10g). Investigation of the contractile function revealed a slight shortening of the contraction time, relaxation time and beating duration of iPSC-CMs in the MM+ES group compared to MM (Fig. 10h), but these changes were smaller compared to those observed with MM+NP+ES versus MM (Fig. 5a). Similar to the MM+NP+ES group, all iPSC-CM cultures in the MM+ES group captured the pacing frequency of 2 Hz (Fig. 10i).

To establish whether ES is the pivotal stimulus for the development of polyploidy of iPSC-CMs, DNA content was measured in iPSC-CMs. The proportion of diploid iPSC-CMs was significantly reduced, while the number of polyploid CMs was significantly increased in the MM+ES group compared to the MM group (Fig. 10j,l). Additionally, we observed the increase in the number of $Ki67^+$ and $Ki67^-$ polyploid iPSC-CMs in the MM+ES group compared to the MM group (Fig. 10k,m,n), similar to those in the MM+NP+ES group (Fig. 8h,i). Taken together, these results demonstrate that ES alone can enhance mitochondrial development, electrophysiological function and polyploidy of iPSC-CMs.”

Fig. 10: Effect of ES on iPSC-CM maturation. **a** Representative morphology of iPSC-CMs under MM and MM+ES conditions. **b** Quantification of cTNT intensity. **c** Quantification of cell volume (FSC-A) and granularity (SSC-A) in the cTNT-positive CMs. **d** Quantification of Tom20 intensity. Log transformation was performed for statistical analysis. **e** A notch event (red arrow) in an iPSC-CM of the MM+ES group. **f** Quantification of AP metrics: RMP, Vmax, APA, and APD₉₀. n = 15 (MM) and 22 (MM+ES) cells from 3-4 independent differentiations of 2 iPSC lines. **g** Statistical analysis of I_{to} . n = 24 (MM) and 25 (MM+ES) cells from 5 independent differentiations of 2 iPSC lines. The stimulation protocol is shown as an inset. **h** Statistical analysis of beating properties: contraction time, relaxation time and beating duration of iPSC-CMs under 0.5 Hz field stimulation (n = 20 cultures/group from 5 independent differentiations of 3 iPSC lines). **i** Heatmap of the ability of iPSC-CMs to adapt to increasing pacing frequencies (n = 9 cultures/group from 3 independent differentiations of 3 iPSC lines). **j,k** Shown are flow cytometry plots for DNA content (**j**) and Ki67 activity (**k**). **l-n** Quantification of diploid and polyploid cells in total iPSC-CMs (**l**) and in Ki67-positive (**m**) and negative (**n**) populations. Statistical analysis

was performed using linear mixed model with Tukey's post-test (**b-d, l-n**), Two-way ANOVA with Sidak's multiple comparison test (**g**), or Kolmogorov-Smirnov test (**f, h**): * $p < 0.05$, ** $p < 0.01$, *** $p < 0.001$, **** $p < 0.0001$. Data are presented as mean \pm SEM (**g**) and in box plots indicating median (middle line), 25th, 75th percentile (box) and min and max data points (whiskers) in **b-d, f, h, l-n**.

3. Nanopatterning on a rigid substrate as used here still prevents symmetric contractions across the substrate and is a major limitation of this technique compared to 3D tissues or 2D cell/tissue patterning techniques that allow consistent fractional shortening across the cells/tissue.

We fully agree with the reviewer about the limitations of nanopatterning on a rigid substrate. To provide more information on the NP surfaces and to explain the limitations of our protocol, we have added the following information in the Methods section (see lines 693-696) and in the Discussion section (lines 535-545).

In the Methods section: "NP coverslips are made of glass coverslips covered with a structured polyethylene glycol diacrylate polymer (800 nm groove width, 800 nm ridge width, and 600 nm height) with a surface stiffness of approximately 7 mPa²⁵."

In the Discussion section: "One of the contributions of the NP surface to the maturation of iPSC-CMs is the improvement of cell alignment, granularity, sarcomere length and contractile behavior, but their structural maturation is still less pronounced compared to 3D tissue. 3D tissue allows symmetric contractions of EHTs with a significantly improved structure, including T-tubule formation and colocalisation with RYR2, sarcomere organization and length^{16,17,18,28}. This difference may be due to the need of 2D monolayers to attach to the surface, where there is a significant mechanical mismatch. The NP surface with a Young's modulus of ~ 7 mPa is much higher compared to diastolic adult human myocardium in the range of 8-15 kPa³⁶ and is therefore still a major limitation for its use in 2D culture when compared to 3D tissue or 2D cultures with other techniques (e.g. on PDMS with 8kPa)²⁴ that allow consistent fractional shortening across the tissue."

4. The methods state that 3 iPSC lines have been used. However, most of the legends report only 2 lines having been used. Moreover, some legends state quite a low number of replicates to be obtained from the number of stated lines/differentiation batches (e.g. 4b legend MM+EP+NP condition only included 9 cells total from 3 differentiation batches each for 2 lines – this would imply each differentiation batch only contributed 1-2 cells) – these sparse numbers could result in inadvertent selection bias. While patch clamping experiments are obviously very time consuming, only data including adequate numbers of cells from multiple batches/lines should be used. Please state specifically the lines used for each experiment. Key results would ideally be shown in 3 lines.

To address this point, we performed additional differentiations to obtain key results from all 3 iPSC lines and to increase the number of replicates. The new data for the key results were integrated into new figures (Figs. 2, 3, 4, 5, 8, 9; Supplementary Fig. 1), including analysis of α -actinin/RYR2 colocalisation; patch clamp measurements (I_{Na} , I_{Ca-L} , I_{to} , I_{K1} , I_{Kr}); contraction analysis (beating duration, contraction time, relaxation time, response to pacing); real-time PCR (*TNNI1*, *TNNI3*, *MYL2*, *MYL7*, *MYH6*, *MYH7*, *OPAI1*, *PPARGC1A*, *PPARA*); and flow cytometry analysis (FSC-A, SCC-A, % of cTNT-positive cells, cTNT intensity, Tom20 intensity, ploidy, Ki67).

In particular, we included data from 3 iPSC lines (isWT7, iWTD2 and iBM76) for I_{Na} , I_{Ca-L} , I_{to} , and I_{K1} , and 2 iPSC lines (isWT7 and iBM76) for I_{Kr} . Two differentiation experiments were performed for each cell line, and $n \geq 3$ cells per differentiation and group were included. Notably, we did not observe significant differences among three iPSC lines with respect to

patch clamp measurements of I_{Na} , I_{Ca-L} , I_{to} , I_{K1} , and I_{Kr} . See Supplementary Fig. 7.

Supplementary Fig. 7: iPSC line-based plotting of patch-clamp measurements of I_{Na} , I_{Ca-L} , I_{to} , I_{K1} , and I_{Kr} . Data from 3 iPSC lines (isWT7, iWTD2 and iBM76) for I_{Na} , I_{Ca-L} , I_{to} , and I_{K1} , and 2 iPSC lines (isWT7 and iBM76) for I_{Kr} . Two differentiation experiments of each cell line were performed. $n \geq 3$ cells per differentiation and group were included. No significant differences among three iPSC lines were observed. Data are presented as mean \pm SD.

5. The number of lines/batches/replicates used for the RNA-seq experiment are not described (at least not in legend or the methods section).

According to the reviewer's comment, we have added the following information to the Methods section, lines 909-910. "iPSC-CMs from three independent differentiations of 2 iPSC lines (iWTD2.1, isWT7.22) were used for RNA sequencing analysis." In addition, the cell line information and culture conditions were included in Supplementary Fig. 5a.

6. Standard deviations rather than SEM should be used to better show the spread of the data.

We appreciate the reviewer's suggestion. In order to optimise the presentation and statistical analysis of the data, we consulted a statistical expert on campus. As a result, we have modified the data plots in the revised manuscript to present the data in box plots indicating median (middle line), 25th, 75th percentile (box) and min and max data points (whiskers) to better show the spread of the data or in lines and symbols as mean \pm SEM.

7. Methods for relative quantification of genes by qRT-PCR are not clear (Fig 4i) – if done by qRT-PCR, primers may have different efficiencies which will markedly affect ratios if directly compared. Referencing each to the housekeeping gene may be more appropriate unless primer efficiencies are directly measured, or the RNA-seq data may be better suited to quantify the relative expression effects.

We thank the reviewer for pointing out this issue. In general, we determine the efficacy of all primers before use through serial dilution of cDNA and the specificity based on gel electrophoresis as well as melting curves. Primer efficacies were calculated using BioRad CFX Manager software. For quantification of gene expression, we used formula 1⁶⁸ to calculate relative expression of each gene to HPRT.

Formula 1:

$$\text{Relative expression} = \frac{1 / (1 + \text{efficacy})^{Ct(\text{target})}}{1 / (1 + \text{efficacy})^{Ct(\text{reference})}}$$

These details were added into the Methods section (lines 902-906) and efficacies were added into Supplementary Table 6.

We also analysed RNA-seq data to quantify the relative expression effects and observed similar results (see the Figure below). Since we have only three conditions in the RNA-Seq analysis, we did not include this data into the revised manuscript.

Relative expression of genes of interest from RNA-seq data, shown as normalised count values or normalised counts ratio. For analysis of differentially expressed genes, counts were fitted to the negative binomial distribution and p values were determined between conditions using the Wald test of DESeq2. * p < 0.05, ** p < 0.01, **** p < 0.0001.

Reviewer #2 (Remarks to the Author):

This is a nice contribution addressing the maturation of human iPSC-CMs with a combination of maturation media (MM), nanopatterned interfaces (NP), and electrical pacing (ES). The experiments are well planned, presenting key data demonstrating the maturation of iPSC-CMs induced by the individual conditions, with the combination of MM + NP + ES being the most impressive. They are expected given the number of publications employing the conditions; however, the detailed genomic and electrophysiology analysis is a step forward for the field. Some minor comments are provided to improve the manuscript.

We thank the reviewer for this positive feedback.

1. More details are needed on the structure of the commercially obtained nanopatterned (NP) coverslips (NanoSurface Coverglass, Curi Bio). What are the essential features of these interfaces? Related, in Fig. 1b it is important to demonstrate cell orientation relative to the NP surface features, i.e. alignment. This should be quantitative, essentially measuring the degree deviation between cells and surface features. What is the difference in the sarcomere structure on NP surfaces, both with and without ES? Essentially, I would like to see data on how the NP surface improves 2D tissue alignment, alters sarcomere structure.

We thank the reviewer for raising this point. To provide more information on the NP surfaces, we have added the following information in the Methods section (see lines 693-696): “NP coverslips are made of glass coverslips covered with a structured polyethylene glycol diacrylate polymer (800 nm groove width, 800 nm ridge width, and 600 nm height) with a surface stiffness of approximately 7 mPa²⁵.”

To study how the NP surface improves cell alignment, we imaged iPSC-CMs stained for α -actinin and nuclei and the NP surface on the coverslip at different z levels using confocal microscopy. The NP direction was defined as 0° in the MM+NP and MM+NP+ES groups, whereas 0° was randomly defined in the B27 and MM groups. The z-disk orientation (stained with α -actinin) and the direction of the elongated nuclei (stained with Hoechst 33342) were calculated with respect to 0° using Cell Profiler software. The information has been included in the Methods section (lines 861-867) and shown in Supplementary Fig. 1a. The quantification of sarcomere alignment based on z-disk orientation and nuclei elongation and sarcomere length have been included in the revised manuscript (lines 99-109):

“Striated pattern of sarcomeres, as demonstrated by the immunostaining against the sarcomeric protein α -actinin, was observed in iPSC-CMs under all conditions (Fig. 1c). Notably, highly organized sarcomeres with well-defined striations, which are aligned into long, continuous myofibrils (stained with phalloidin) that run the length of the cell, were observed only in the MM+NP and MM+NP+ES groups (Fig. 1c,d). Quantitative analysis revealed that the majority of iPSC-CMs in the MM+NP and MM+NP+ES groups showed sarcomere patterns at an angle of 90° to the NP direction with elongated nuclei along the NP direction, whereas iPSC-CMs in the B27 and MM groups revealed random orientations of sarcomeres and nuclei in all directions (Fig. 1c,e,f). In addition, the sarcomere length is significantly longer in iPSC-CMs in the MM+NP and MM+NP+ES groups compared to those in the B27 and MM groups (Supplementary Fig. 1b).”

and in new Fig. 1 and Supplementary Fig. 1b.

Fig. 1: Study design and structural characterisation of iPSC-CMs. **a** Schematic overview of the study design. Differentiated iPSC-CMs were digested on day 15-16 (d15-16) and randomly divided into 4 experimental groups to investigate the effects of MM, NP and ES. Extensive characterisation of the cells was performed on day 42 (d42). For some experiments, including calcium imaging, seahorse assays and MEA measurements, iPSC-CMs were replated into the corresponding assay plates on d42 and allowed to recover for another 7 days. **b** Representative morphology of iPSC-CMs under different conditions at d42. Scale bar, 200 μm for all four groups. **c** Representative immunostaining for α -actinin, RYR2 and Hoechst33342. **d** Representative immunostaining for Cx43, phalloidin and Hoechst33342. **e,f** Quantification of sarcomere alignment based on z-disk orientation (**e**) and

nuclei elongation (f). The method used to analyse the sarcomere alignment is shown in Supplementary Fig. 1a. Data are presented as mean and SD from averages of each experiment (2 independent experiments with 2 iPSC lines and with $n > 100$ nuclei and $n > 15,000$ z-disks per experiment).

Supplementary Fig. 1: Analysis NP orientation, sarcomere length and α -actinin-RYR2 colocalisation. **a** Detection of iPSC-CMs stained for α -actinin and nuclei and the NP on the coverslip at different z levels using confocal microscopy. Orthogonal projections of two images illustrating the z level of the NP surface (left) and the cell (right). Yellow lines and arrows indicate the position of the projection. Red arrows mark the grooves of the NP surface (left, y - z projection), and blue arrows mark the z -disks (right, x - z projection). To analyse sarcomere alignment, the NP direction was defined as 0° in the MM+NP and MM+NP+ES groups, whereas 0° was randomly defined in the B27 and MM groups. The z -disk orientation (stained with α -actinin) and the direction of the elongated nuclei (stained with Hoechst 33342) were calculated with respect to 0° using Cell Profiler software. **b** Sarcomere length of iPSC-CMs. $n = 30$ cells/group from 3 iPSC lines (10 cells/line). **c** Quantification of α -actinin/RYR2 colocalisation. $n = 18$ images/group from 3 iPSC lines (6 images/line). Statistical analysis was performed using Kruskal-Wallis test, * $p < 0.05$, ** $p < 0.01$. Data are presented as box plots indicating median (middle line), 25th, 75th percentile (box) and min and max data points (whiskers) in **b** and **c**.

2. “However, it is worth noting that 3D-tissue generation is challenging and the experimental throughput is lower compared to 2D-cultures.” Although this statement is true, it ignores the comprehensive data generated showing 2D surfaces are inferior to 3D microtissues. Refs. 16 & 17 demonstrated this, among other papers, including Ronaldson-Bouchard (which was omitted - Ronaldson-Bouchard, K., Ma, S.P., Yeager, K. et al. Advanced maturation of human cardiac tissue grown from pluripotent stem cells. *Nature* 556, 239–243 (2018). <https://doi.org/10.1038/s41586-018-0016-3>). One main reason for this is that the 2D monolayers must be attached to the surface, where there is a significant mechanical mismatch, thus altering contraction behavior. I suggest the authors address the 2D vs 3D issue in the context of improved alignment, in part supported by the recommendations in #1.

Following the reviewer's suggestion, we have included the following discussion in the Discussion section (lines 535-545): "One of the contributions of the NP surface to the maturation of iPSC-CMs is the improvement of cell alignment, granularity, sarcomere length and contractile behavior, but their structural maturation is still less pronounced compared to 3D tissue. 3D tissue allows symmetric contractions of EHTs with a significantly improved structure, including T-tubule formation and colocalisation with RYR2, sarcomere organization and length^{16,17,18,28}. This difference may be due to the need of 2D monolayers to attach to the surface, where there is a significant mechanical mismatch. The NP surface with a Young's modulus of ~7 mPa is much higher compared to diastolic adult human myocardium in the range of 8-15 kPa³⁶ and is therefore still a major limitation for its use in 2D culture when compared to 3D tissue or 2D cultures with other techniques (e.g. on PDMS with 8kPa)²⁴ that allow consistent fractional shortening across the tissue."

3. Present calculated IC₅₀ or EC₅₀ from the drug experiments. How do these data compare to other studies using iPSC-CMs using maturation media and pacing? How do they compare to human heart tissue slices.

According to the reviewer's comment, we have added the following information in the Discussion section (lines 608-613): "The EC₅₀ values for isoprenaline chronotropy in this study fell within the nanomolar range (1-47 nM) reported for EHTs cultured in MM^{16,22}. Slightly higher EC₅₀ values have been reported for EHTs subjected to pacing (98 nM)²⁸, and for fetal cardiac tissue (30 nM)²⁸ and adult human heart slices (180 nM)⁵². Notably, significant differences in EC₅₀ values for isoprenaline chronotropy were observed in two different iPSC lines (1 nM vs. 47 nM)¹⁶."

For the drug response data of verapamil and E-4031 we decided to not determine IC₅₀/EC₅₀ values as drug concentrations have not been escalated until all spontaneous activity ceased in our study, and therefore calculation of a reliable IC₅₀/EC₅₀ is not possible. We have added the following in the discussion (lines 615-619): "Several studies reported that verapamil inhibits the beating activity of iPSC-CMs^{15,16,53}, probably because depolarisation in immature iPSC-CMs does not rely exclusively on *I_{Na}*, as in adult CMs, but also on *I_{Ca-L}*¹⁵. In line with these studies^{15,16}, our data demonstrate that increased maturation reduces verapamil-induced beating arrest of iPSC-CMs."

4. Were any arrhythmia-like events observed, and if so, were there differences between the treatments.

We thank the reviewer for pointing out this important issue. In general, we did not observe systematically appearance of arrhythmias between the treatments. In the Methods section (lines 763-767), we have included: "In addition, we quantified the number of quiescent and arrhythmic cultures after treatment with verapamil, E-4031 and isoprenaline based on beating rate variation using Cardiac Analysis Tool.". We have included the data in the revised manuscript (lines 315-317):

"We observed beating arrest in cultures from the B27 (17/17), MM (6/17) and MM+NP (6/18) groups at 1 μM verapamil, but no significant induction of arrhythmias with all three substances (Supplementary Fig. 4c,d)." and as Supplementary Fig. 4c,d

c Number of quiescent cultures

	Verapamil				E-4031				Isoprenaline			
	B27	MM	MM+NP	MM+NP+ES	B27	MM	MM+NP	MM+NP+ES	B27	MM	MM+NP	MM+NP+ES
Cultures analysed (n wells/N batches)	17/3	17/3	18/3	18/3	10/2	12/2	11/2	12/2	18/3	23/4	23/4	24/4
Baseline	0	0	0	0	0	0	0	0	0	0	0	0
1 nM	0	0	0	0	0	0	0	0	0	0	0	0
10 nM	0	0	0	0	0	0	0	0	0	0	0	0
100 nM	5	0	0	0	n.d.	n.d.	n.d.	n.d.	0	0	0	0
1 μ M	17	6	6	0	n.d.	n.d.	n.d.	n.d.	0	0	0	0

d Number of arrhythmic cultures

	Verapamil				E-4031				Isoprenaline			
	B27	MM	MM+NP	MM+NP+ES	B27	MM	MM+NP	MM+NP+ES	B27	MM	MM+NP	MM+NP+ES
Baseline	0	0	0	0	0	0	0	0	0	0	0	0
1 nM	2	1	0	1	0	0	2	1	0	0	1	2
10 nM	1	0	2	0	0	2	2	1	0	0	0	0
100 nM	0	0	0	1	n.d.	n.d.	n.d.	n.d.	0	0	0	1
1 μ M	--	3	0	0	n.d.	n.d.	n.d.	n.d.	0	0	0	0

Supplementary Fig. 4: Drug response of iPSC-CMs using MEA. a Experimental scheme. **b** Sequential drug adding protocol after baseline recording. **c,d** Number of quiescence (**c**) and arrhythmic (**d**) cultures from $n = 17-18$ cultures/group from 3 independent experiments of 2 iPSC lines for verapamil; $n = 10-12$ cultures/group from 2 independent experiments of 2 iPSC lines for E-4031; and $n = 18-24$ cultures/group from 3-4 independent experiments of 2 iPSC lines for isoprenaline. n.d., not done. **e** Representative field potential traces illustrating the change in spike amplitude induced by verapamil.

5. The discussion does not compare their results to previously published work, especially those that use similar maturation media.

In the revised manuscript, we have included the following in the Discussion section (lines 522-534): “The core of the maturation strategy is the application of FA-supplemented MM in iPSC-CMs, which has been reported in several studies^{13,16,20,22}. Previous studies have shown that MM induces the metabolic transition from glucose-based energy production to FA β -oxidation^{13,20} and changes in the expression of genes associated with calcium cycling, ion channels and structural proteins^{13,16,22}. This metabolic transition is essential for increased Ca^{2+} transient kinetics, I_{K1} density, and iPSC-CM hypertrophy and improved AP parameters, mitochondrial density and function, contractility and drug response not only in 2D-cultures^{13,20} but also in 3D-tissues generated with iPSC-CMs and fibroblasts^{16,22}. In line with these studies, we show here that MM induces enhanced structural (cell size, cTNT and Tom20 intensity), electrophysiological (increased I_{Na} , I_{to} , I_{K1} and I_{Kr} density, improved FP parameters and Ca^{2+} cycling) and metabolic (mitochondrial respiration) maturation of iPSC-CMs, leading to improved contractility and drug response. The effect of MM on I_{Na} , I_{Ca-L} and Ca^{2+} cycling is largely due to the high concentration of calcium in MM.”.

Lines 608-613: “The EC_{50} values for isoprenaline chronotropy in this study fell within the nanomolar range (1-47 nM) of EC_{50} values reported for EHTs cultured in MM^{16,22}. Slightly higher EC_{50} values have been reported for EHTs subjected to pacing (98 nM)²⁸, and for fetal cardiac tissue (30 nM)²⁸ and adult human heart slices (180 nM)⁵². Notably, significant differences in EC_{50} values for isoprenaline chronotropy were observed in two different iPSC lines (1 nM vs. 47 nM)¹⁶.”

Minor

Fig. 1c – The APD90 is very long and not physiological for all the conditions, except MM+NP+ES. This is a departure from many studies in 2D showing ADP in a more physiological range, ~ 400-500 ms.

The absolute values of AP parameters are associated with various measurement conditions, such as recovery time after trypsinization, intracellular and extracellular solutions, and recording temperature. In most studies published using human iPSC-CMs, recovery time after trypsinization is about 1 week, however, in our study, we used the cells directly after overnight recovery in cardio-digestion medium in order to minimise the time that CMs from the MM+NP and MM+NP+ES groups spent in non-NP and non-ES conditions. In addition, we performed AP recordings at room temperature. The information was given in the Methods section (lines 726-732): “For action potential (AP) recordings and I_{Kr} measurements using manual patch clamp technique, CMs from all four groups were used directly after overnight recovery in cardio-digestion medium in order to minimise the time that CMs from the MM+NP and MM+NP+ES groups spent in non-NP and non-ES conditions. The pipette and extracellular solutions used for AP and I_{Kr} recordings are listed in Supplementary Table 4. All manual patch clamp experiments were performed at RT using a ruptured whole-cell patch clamp with a HEKA EPC10 amplifier and Patchmaster (HEKA Elektronik).”.

Fig. 3d – pacing inset difficult to understand.

We have included explanations to pacing inset in the figure legend. For example, for the I_{K1} recording (Fig. 3d): “The holding potential was set at -40 mV. I_{K1} was recorded by increasing the test potential from -130 mV to +10 mV in 10 mV steps. Each pulse lasted for 2 s and the

sweep interval was 10 s. The protocol was then repeated in the presence of 0.5 mM BaCl₂ and the Ba²⁺-sensitive current was calculated as I_{KI} .”

429 – line “ These findings align with previously reported data on the positive effects of FA supplementation^{13,20}. Refs 16 and Mills’ PNAS 2017 paper should be cited here, probably others as well. (<https://www.pnas.org/doi/full/10.1073/pnas.1707316114>)

We have included the mentioned publications and discussed those papers together with our findings (lines 522-534).

Reviewer #3 (Remarks to the Author):

The authors submit a manuscript investigating the role of patterning, metabolic substrate and electrical pacing on maturation. Overall the authors present some interesting findings relating to differences in morphology, electrophysiology, transcriptomes, drug responses and cell cycle. This is interesting for the field whereby researchers are trying to further mature hPSC-CM for accurate modelling. Overall the study seems well executed and written, although, there are a number of concerns that need to be addressed in order to determine whether their findings are impactful. Furthermore, for a journal of Nature Communications standing, it would be of substantial importance to validate modelling of an important disease/drug/phenotype that can be uncovered by their more mature system over regular hPSC-CM culture. Without such a demonstration the speculation that these results can be extrapolated into eg patient specific modelling are currently unsubstantiated.

We thank the reviewer for the feedback.

Major

1. Others have seen better responses in 2D hPSC-CM to drugs in comparison to the B27 control group (eg. PMID: 28153730). Therefore, this reviewer wonders whether this group in the paper is especially immature and how significant the maturation in the other groups actually is?

A strength of our study is that we systematically compared iPSC-CMs in four different conditions in terms of their electrophysiological properties and drug responses, and clearly demonstrate that the drug responses of iPSC-CMs are improved when the cells are at a higher level of electrophysiological maturation. In our study, iPSC-CMs in the B27 group are less mature than the other three groups. They showed a loss of beating activity when treated with 1 μ M verapamil. Similar results have been reported in other studies (PMID: 24030418; PMID: 35478228). For example, the treatment of iPSC-CMs with 1 μ M verapamil caused half of the cultures to stop beating in 82-day-old cultures (PMID: 24030418). Spontaneous beating was four times more sensitive to verapamil at earlier stages of differentiation (half-maximal inhibitory concentration, or IC_{50} , at 82 days post-induction 410.65 ± 40.80 nM vs. 103.20 ± 6.03 in 30-day-old cultures). In our opinion, the different iPSC lines, culture conditions, differentiation time, and electrophysiological platforms used for maturation analysis make it difficult to directly compare the maturation levels of our cultures with others. For example, Huebsch et al. (PMID: 35478228) reported different sensitivities to verapamil in two different iPSC lines in the same B27 medium (IC_{50} : 90 nM vs. 1000 nM) and showed significant differences in EC_{50} values for isoprenaline chronotropy (1 nM vs. 47 nM).

2. For the n numbers were approximately the same n numbers obtained for both B27 controls and the different conditions within each experiment before pooling of the experiments? If different for each experiment, the results will be subject to potential batch effects which may not be appropriately controlled for.

We agree with the reviewer that significant differences in n numbers for each group and experiment may lead to the results being subjected to potential batch effects. To avoid this problem, we carefully controlled the n numbers for each group. For example, for sarcomere

alignment, colocalisation, MEA, contractile, calcium transient, flow cytometry, seahorse, and real-time PCR, we used batch-matched sample size for each condition. For patch clamp experiments, which are very time-consuming, we included data from 3 iPSC lines for I_{Na} , I_{Ca-L} , I_{to} , and I_{K1} , and 2 iPSC lines (isWT7 and iBM76) for AP and I_{Kr} , two differentiation experiments were performed for each cell line, and $n \geq 3$ cells per differentiation and group were included. Notably, we did not observe significant differences among three iPSC lines with respect to patch clamp measurements of I_{Na} , I_{Ca-L} , I_{to} , I_{K1} , and I_{Kr} . See Supplementary Fig. 7.

Supplementary Fig. 7: iPSC line-based plotting of patch-clamp measurements of I_{Na} , I_{Ca-L} , I_{to} , I_{K1} , and I_{Kr} . Data from 3 iPSC lines (isWT7, iWTD2 and iBM76) for I_{Na} , I_{Ca-L} , I_{to} , and I_{K1} , and 2 iPSC lines (isWT7 and iBM76)

for I_{Kr} . Two differentiation experiments of each cell line were performed. $n \geq 3$ cells per differentiation and group were included. No significant differences among three iPSC lines were observed. Data are presented as mean \pm SD.

3. I am not sure the appropriate statistical tests were performed as some data looks skewed and some do not seem to have equal standard deviations. That is of considerable concern as I am not sure some of the comparisons are actually statistically different if the appropriate tests are used.

We thank the reviewer for pointing out this issue. We consulted a statistical expert on campus and performed new statistical analysis accordingly throughout the manuscript and included a description of the statistical analysis in the Methods section (lines 968-978) and in the figure legends.

“Statistical analysis was conducted using R Studio by applying a linear mixed model (lmer, package lmerTest), considering experimental groups and cell lines as fixed variables and independent batches as a random variable. Residuals were evaluated for normal distribution based on plots of residuals vs. fitted values and Shapiro-Wilk test (check_normality function), and for homogeneity with Bartlett’s test (check_homogeneity function). Significance between groups was calculated with emmeans function, applying Tukey’s post-test to adjust for multiple comparisons. If the mixed model could not be applied due to violation of normality or homogeneity, statistical analysis was performed using Kruskal-Wallis test with Dunn’s correction for multiple comparisons (comparison of > 2 groups), Kolmogorov-Smirnov (comparison of 2 groups), or two-way ANOVA with Sidak’s post-test for multiple comparisons using GraphPad Prism 10, indicated in figure legends.”.

4. Figure 1G-I - Cells are flat and pancake-like rather than aligned like mature CM. The representative section of a small area of the sarcomeres is misleading. With the variability in Fig1H quantification its difficult to determine whether the changes are biologically meaningful. Also the CX43 staining is adjacent rather than end-end in Fig1I.

We thank the reviewer for pointing out this issue. The imaging in old Fig. 1g-h were from cells that were digested at day 42 and replated on glass coverslips for immunostaining. Therefore, cells were not aligned as on the NP surface shown in Fig. 1b. We agree that this was misleading. Therefore, we have performed new experiments to stain iPSC-CMs on the NP surface. Representative images of cells stained for α -actinin and RYR2 as well as Cx43 and phalloidin have been included in the revised Fig. 1c and d, respectively.

Following the suggestion from reviewer 2, to study how the NP surface improves cell alignment, we imaged iPSC-CMs stained for α -actinin and nuclei and the NP surface on the coverslip at different z levels using confocal microscopy. The NP direction was defined as 0° in the MM+NP and MM+NP+ES groups, whereas 0° was randomly defined in the B27 and MM groups. The z-disk orientation (stained with α -actinin) and the direction of the elongated nuclei (stained with Hoechst 33342) were calculated with respect to 0° using Cell Profiler software. The information has been included in the Methods section (lines 861-867) and shown in Supplementary Fig. 1a. The quantification of sarcomere alignment based on z-disk orientation and nuclei elongation and sarcomere length have been included in the revised manuscript (lines 99-107) and in new Fig. 1 and Supplementary Fig. 1b.

Regarding to Cx43 staining, we have rephrased the description (lines 114-117): “In the B27 and MM groups, the gap junction protein connexin 43 (Cx43) was partially localised to perinuclear regions and in the cytosol, whereas Cx43 membrane localisation was increased in CMs of the MM+NP and MM+NP+ES groups (Fig. 1d).”

Fig. 1: Study design and structural characterisation of iPSC-CMs. **a** Schematic overview of the study design. Differentiated iPSC-CMs were digested on day 15-16 (d15-16) and randomly divided into 4 experimental groups to investigate the effects of MM, NP and ES. Extensive characterisation of the cells was performed on day 42

(d42). For some experiments, including calcium imaging, seahorse assays and MEA measurements, iPSC-CMs were replated into the corresponding assay plates on d42 and allowed to recover for another 7 days. **b** Representative morphology of iPSC-CMs under different conditions at d42. Scale bar, 200 μm for all four groups. **c** Representative immunostaining for α -actinin, RYR2 and Hoechst33342. **d** Representative immunostaining for Cx43, phalloidin and Hoechst33342. **e,f** Quantification of sarcomere alignment based on z-disk orientation (**e**) and nuclei elongation (**f**). The method used to analyse the sarcomere alignment is shown in Supplementary Fig. 1a. Data are presented as mean and SD from averages of each experiment (2 independent experiments with 2 iPSC lines and with $n > 100$ nuclei and $n > 15,000$ z-disks per experiment).

5. Reduction of proliferation occurs with maturation condition in particular fatty acids-exercise (PMID: 28916735). This would decrease cell number and increase cell size, rather than a direct impact of eg. hypertrophy. This explanation for the changes is highlighted by the cell cycle analysis in Fig7, and therefore the likely mechanism of these changes.

It is true that a reduction in proliferation is correlated with a decrease in cell number, but this is not always the case with an increase in cell size. In 2D culture, a decrease in cell number may provide cells more space to expand, in which the cell surface area may be increased. However, we measured cell volume and granularity using flow cytometry, suggesting an increase in protein synthesis in iPSC-CMs especially in the MM+NP+ES group compared to the B27 group. Therefore, we believe that iPSC-CMs undergo hypertrophic growth.

6. For the gene expression analyses the genes in the heatmaps seem to be cherry picked, and given the variability in the expression for many of them its difficult to determine which are and are not regulated. It would be preferable to only present and highlight regulated genes. Additionally, the gene expression could be benchmarked to human heart tissue so that the reader can understand what the remaining deficits in maturity are.

Indeed, the genes in the heatmaps were selected to show not only all significantly regulated genes, but also a set of genes relevant for a specific context. We agree that this may lead to uncertainty which genes are significantly regulated, therefore, in the revised manuscript, we have provided the statistical significance for the comparisons of all 3 groups in Fig. 7g, Fig. 8b, and Fig. 9b,c,h, and explained in the figure legends.

In the Discussion section (lines 645-649), we have included the following discussion: “Finally, when we compared the expression pattern of marker genes, important for ventricular CM maturity, cardiac ion channels, and genes related to cell cycle activity and mitochondrial development in iPSC-CMs of the MM+NP+ES group with published RNA-seq datasets of human fetal ventricle and adult heart samples, we found that the maturity level of iPSC-CMs was intermediate between that of human fetal and adult CMs (Supplementary Fig. 6).”

The following paragraph have been included in the Methods section (lines 952-963): “To benchmark the gene expression pattern of iPSC-CMs in the MM+NP+ES group with human heart tissue, our own data and published bulk RNA-seq datasets (GSE62913) from fetal ventricles (GSM1536186, GSM1536187) and adult heart samples (GSM1536192, GSM1536193) have been were aligned using STAR to the homo sapiens reference hg38 and using Ensembl annotation 104. A list of 210 genes, including marker genes for ventricular CMs, ion channels, and genes related to cell cycle activity and mitochondrial development that were affected by ES in our study, was used for comparison. Fragments were counted using FeatureCounts to assign fragments to gene features. The counts table was processed using R and the libraries DESeq2 (v1.38.3) and edgeR (v3.40.2). DESeq2 standard methods were used

to normalise the data. Edge methods were applied to the normalised data to correct for possible effects of different library preparation methods. The normalised and corrected data were then visualised using the R library ggplot2.”

Supplementary Fig. 6: RNA sequencing analysis. **a** Principal component analysis of MM+NP+ES group in comparison to published RNA-seq datasets from human fetal ventricle (HFV) and human adult heart (HAH) samples, based on 210 manually selected genes including CM markers, ion channels as well as SRF target genes, mitochondrial genes and cell cycle genes that were regulated in MM+NP+ES group in comparison to MM (gene list and dataset see Supplementary Data 3). **b** Heatmap of all 210 genes. **c** Heatmaps of DEGs ($p < 0.05$) between MM+NP+ES and HFV, grouped into categories related to main manuscript.

7. The qPCR results seem to be highly variable. Are the differences also seen in the RNA-seq? What do the data points represent? Are they individual wells, were wells scraped and pooled together? Expanding on the details of sample preparation in the first part of the ‘Real-time PCR’ methods section would also help (approximate quantity of cells, volume of TRIzol, and chloroform etc)

We thank the reviewer for pointing out this issue. We analysed RNA-seq data to quantify the relative expression effects and observed similar results (see the Figure below). Since we have only three conditions in the RNA-Seq analysis, we did not include this data into the revised manuscript.

Relative expression of genes of interest from RNA-seq data, shown as normalized count values or normalized counts ratio. For analysis of differentially expressed genes, counts were fitted to the negative binomial distribution and p values were determined between conditions using the Wald test of DESeq2. * p < 0.05, ** p < 0.01, **** p < 0.0001.

In the Methods section (lines 882-892), detailed information on sample preparation for ‘Real-time PCR’ has been included: “On day 42, the cells (1×10^6 cells/well) were washed with ice-cold PBS and scraped from the culture plate or NP coverslips. Two wells of each condition were pooled as one sample and centrifuged at 2,000g at 4°C. All steps were performed on ice. Cell pellets were lysed in 1 mL TRIzol™ (Thermo Fisher Scientific) and homogenized through continuous pipetting (30x) using a metal syringe. Lysates were centrifuged for 5 min at 12,000g at 4°C, the clear supernatant was transferred into a new tube, and 0.2 mL chloroform was added. Samples were mixed thoroughly, incubated for 3 min on ice and centrifuged for 15 min at 4°C and 15,000g to separate phases. The upper phase containing the isolated RNA was carefully collected and mixed with doubled volume of 95% ethanol. The solution was loaded on RNA columns of RNeasy isolation kit (Qiagen), RNA was purified according to manufacturer’s protocol, and eluted in RNase-free water.”.

8. The observation of Cx43 in the perinuclear region is interesting. GJA5 is translated by ribosomes in this region to generate Cx43, which is then trafficked to the cell membrane (PMID: 28576298). Could trafficking of proteins also be another mechanism that is being improved in the MM + NP and MM + NP + ES groups? Does the transcriptomics data provide any insight?

We agree with the reviewer that Cx43 trafficking from the perinuclear region and cytosol to the cell membrane is a very interesting topic. It is possible that improved trafficking of proteins important for the formation of Cx43 gap junction plaques at intercalated discs is another mechanism that is being improved in the MM + NP and MM + NP + ES groups, particularly in the MM+NP+ES group. We analysed the expression of some genes at intercalated discs, including *MAPRE1/2/3*, *CDH2*, *DCTN1*, *DSP*, *PKP2*, *JUP* and *NEDD4* in our RNA-seq data

(see below). We found that *CDH2* and *DSP* were significantly upregulated in MM+NP+ES compared to MM, which may indicate increased cell-cell contacts and intercalated disc formation. However, the expression of *MAPRE1* and *PKP2* was decreased in MM+NP+ES. Further studies on protein levels and protein trafficking are needed to confirm this hypothesis. As this is not the focus of this manuscript, we have not included the RNA-seq analysis for these genes, but included a short discussion in the Discussion section (lines 640-644): “Future studies should focus on whether/how the downregulation of MAPK/PI3K-AKT signalling in iPSC-CMs regulates the gene expression related to ion channel (for example, K⁺ and Na⁺ channels) maturation and function⁶⁰ as well as the formation of Cx43 gap junction plaques⁶¹ and intercalated discs^{43,62}.”.

Minor

9. Use of colour on graphs. Appreciate the consistent use of colours throughout the paper to separate the different groups that were tested, it greatly enhances readability and interpretability. However, in some graphs the formatting choices obscure the data. The best example I can see is Fig 5g. The boxplots with error-bars and individual data points without the use of semi-transparency should be avoided.

Following the reviewer’s suggestions, we have replaced individual data points with transparent symbols to improve readability in all figures.

10. Lines 62-66. Could add mention of abundance, localization, lack or presence of t-tubules, as additional differences between mature cardiomyocytes and iPSC-CMs. Or add reference to reviews where that is discussed.

Following the reviewer’s suggestions, we have included additional differences in this part (lines 62-64): “...and calcium handling, as well as less structural maturity (e.g. fewer mitochondrial networks, lack of T-tubules and intercalated discs)^{1,9} in iPSC-CMs compared to adult CMs.”.

11. Lines 109-112. Figure 1g-h. The authors statements that “the α -actinin/RYR2 co-localization was enhanced in both the MM and MM + NP groups...” is not supported by the statistical comparisons in Figure 1h. Only in the MM + NP + ES group was there a statistically significant increase in co-localization.

We thank the reviewer for pointing out this issue. We have rephrased this sentence (lines 113-114): “The α -actinin/RYR2 colocalisation was significantly enhanced in the MM+NP+ES group compared to the B27 and MM groups (Fig. 1c, Supplementary Fig. 1c).”.

12. Line 141. The “(43%)” was confusing

We have rephrased this sentence (lines 135-137): “We observed 43% of iPSC-CMs in the MM+NP+ES group with a ‘notch-and-dome’ AP morphology, which was not seen in the other groups (Fig. 2a).”.

13. Line 163-164. Fig 2a. The “representative” trace chosen for the MM+NP+ES group has a RMP of about -75mV (it is difficult to tell due to the spacing of the ticks on the y-axis), but the average RMP for the same group shown in Fig 1b is about -65mV.

We thank the reviewer for pointing this out. RMP is defined by analysing at least 5 consecutive stable APs using LabChart 8 software (ADInstruments). The trace shown in Fig. 2a for the MM+NP+ES group has an RMP of -69.02 mV, which represents the median of the cells in the group. As mentioned above, to better show the spread of the data, we have modified the data plots in the revised manuscript to present the data in box plots indicating median (middle line), 25th, 75th percentile (box) and min and max data points (whiskers) (see new Fig. 2b).

14. Line 163. A summary table of in the extended data describing all of the parameters of these experiments will enhance the interpretation of the data and could be a valuable resource for others doing similar work.

We thank the reviewer for pointing out this issue. In the revised manuscript, we have included two summary tables (Supplementary Tables 2 and 3) describing the AP and FP parameters.

Supplementary Table 2: Action potential parameters in the four groups. Data are presented as mean \pm SD

	RMP (mV)	Vmax (V/s)	APA (mV)	APD ₉₀ (ms)
B27	-44.1 \pm 9.8 (n=21)	4.2 \pm 1.4 (n=21)	70.6 \pm 16.7 (n=21)	980.5 \pm 393.1 (n=13)
MM	-49.7 \pm 8.5 (n=25)	5.0 \pm 1.1 (n=25)	78.7 \pm 13.9 (n=25)	732.3 \pm 256.5 (n=24)
MM+NP	-58.2 \pm 7.4 (n=21)	6.6 \pm 2.5 (n=21)	88.4 \pm 10.8 (n=21)	752.5 \pm 375.7 (n=22)
MM+NP+ES	-65.6 \pm 8.5 (n=17)	11.0 \pm 7.4 (n=14)	97.8 \pm 12.7 (n=17)	539.3 \pm 186.2 (n=20)

Supplementary Table 3: Field potential parameters in the four groups. Data are presented as mean \pm SD

	CV (cm/s)	Spike amplitude (mV)	Spike slope (V/s)
B27	12.5 \pm 5.8 (n=24)	3.2 \pm 1.2 (n=24)	-3.6 \pm 1.8 (n=24)
MM	22.3 \pm 3.7 (n=24)	10.4 \pm 4.9 (n=24)	-21.2 \pm 11.3 (n=24)
MM+NP	25.6 \pm 4.3 (n=24)	10.7 \pm 4.8 (n=24)	-23.6 \pm 11.1 (n=24)
MM+NP+ES	27.8 \pm 7.3 (n=24)	13.6 \pm 5.3 (n=24)	-29.1 \pm 12.6 (n=24)

15. Fig 4d. The units appear to be the measured Fura-2 ratio from Fig 4c, for clarity can this be added to the y-axes or to the figure legend.

In the revised manuscript, we have included Fura-2 ratio to the y-axes in Fig. 4d.

16. Line 282 and Fig 5h. Inconsistent use of the name for isoprenaline/isoproterenol. It is best to pick one and stick to it throughout the manuscript to prevent any confusion for non-experts.

We thank the reviewer for this comment. We have used isoprenaline consistently throughout the revised manuscript.

17. Line 318. The authors refer to a “transcription factor target database.” It is not obvious what this is to this reviewer, can more details be provided?

Lines 365-367 we have added the web link for the TFT collection. In the Methods section (lines 940-945), we have included detailed information: “Gene set enrichment analysis (GSEA) was performed based on normalised count data using GSEA software v.4.3.2⁶⁹ and canonical pathways (CP) collection (C2, CP: c2.cp.v2023.2.Hs, <https://www.gsea-msigdb.org/gsea/msigdb/human/genesets.jsp?collection=CP>) or TFT collection (C3, TFT: c3.tft.v2023.2, <https://www.gsea-msigdb.org/gsea/msigdb/human/genesets.jsp?collection=TFT>), two Human Molecular Signatures Database (MSigDB) collections provided by the Broad institute”.

18. Line 394. T-type calcium channels are upregulated, can the authors provide comment in the discussion on the potential impact of this on automaticity of the iPSC-CMs in context of the other changes that were seen? Perhaps also include mention of HCN1 and HCN4 which are known to be more abundant in iPSC-CMs than (ventricular) adult cardiomyocytes (PMID: 37028405).

We thank the reviewer for pointing out this important issue. We have analysed our RNA-seq data and found that the expression of HCN1 and HCN4 were not altered in the MM+NP+ES group compared to the MM group, whereas sinoatrial node-specific genes SHOX2 and RGS6 were significantly downregulated (Supplementary Data 2). In the Discussion section (lines 628-631), we have included the following information: “Future studies should investigate how the automaticity of iPSC-CMs in the MM+NP+ES is affected, which is controlled by the coupled system of Ca²⁺ and membrane clocks⁵⁵ and may involve T-type calcium channels and HCN channels⁵⁶.”.

19. Fig 8h. “SCL8A1” should be “SLC8A1”?

The typo has been corrected.

POINT-BY-POINT RESPONSE TO REVIEWERS' COMMENTS

Reviewer #1 (Remarks to the Author):

The authors have comprehensively addressed my critiques. They have added new experiments to add a comparison for MM without NP, comparison of increased media calcium concentration alone (via DMEM), and increased replicates. These additions make the manuscript much stronger to support the conclusions. As noted in my initial review, many of the findings have been previously reported, but the investigators here have provided more electrophysiologic measurements, which will be a nice addition to the iPSC-CM maturation field. I do think it would be very worthwhile to work the major effect of calcium and MM-ES (with minor or no effect of NP) in electrophysiologic maturation into both the abstract and introduction since the summaries currently still make it appear as though the NP is important to a combinatorial effect. In particular, calcium should be highlighted given the investigators focus on electrophysiology and the fact that calcium seemed to be the major driver of some of the EP parameters (this is has not been emphasized much in the existing literature).

We thank the reviewer for the thoughtful feedback.

Following the Reviewer's suggestion, we have revised the abstract and the introduction to more accurately reflect the findings. Specifically, we have highlighted the distinct contributions of ES, calcium and NP to the structural, metabolic and electrophysiological maturation of iPSC-CMs. In particular, we have emphasized calcium's pivotal role in driving electrophysiological maturation of iPSC-CMs. Please see the Abstract, lines 32-43. "Here, we demonstrate an approach that combines lipid-enriched maturation medium with a high concentration of calcium, nanopatterning of culture surfaces and electrostimulation to generate iPSC-CMs with advanced electrophysiological, structural and metabolic phenotypes. Systematic testing reveals that electrostimulation is the key driver of enhanced mitochondrial development and metabolic maturation and improved electrophysiological properties of iPSC-CMs. Increased calcium concentration strongly promotes electrophysiological maturation, while nanopatterning primarily facilitates sarcomere organisation with minor effect on electrophysiological properties. Transcriptome analysis reveals that activation of HMCEs and TFAM targets contributes to mitochondrial development, whereas downregulation of MAPK/PI3K and SRF targets is associated with iPSC-CM polyploidy. These findings provide mechanistic insights into iPSC-CM maturation, paving the way for pharmacological responses that more closely resemble those of adult CMs."

In the Introduction section, lines 79-90, "Here, we systematically examined the effects of FA-enriched maturation medium (MM), increased calcium concentrations in the medium, NP and ES on the electrophysiological, structural and metabolic maturation of iPSC-CMs through comprehensive functional and molecular analyses. While NP primarily promoted structural maturation with limited impact on electrophysiological properties, elevated Ca^{2+} concentrations in the medium strongly influenced the electrophysiological characteristics of iPSC-CMs. Notably, ES emerged as the key driver of enhanced mitochondrial development and metabolic maturation and improved electrophysiological properties. The combined application of MM with a high calcium concentration, NP and ES led to significant changes in

the sensitivity of iPSC-CMs to cardioactive drugs, yielding pharmacological responses that more closely resemble those of adult CMs. Our data provide mechanistic insights into the distinct roles of these stimuli in shaping the cellular structure, metabolism and electrophysiology of iPSC-CMs.”

Reviewer #3 (Remarks to the Author):

The authors have addressed all of my comments.

We thank the reviewer for this positive feedback.

Reviewer #4 (Remarks to the Author):

This manuscript defines a rigorous study to evaluate separate and combined effects of methods for driving cardiomyocyte maturation. I feel the authors have responded appropriately to all reviewer comments and have greatly improved the manuscript, particularly by consulting with a statistician and clearly evaluating and explaining the appropriate statistical tests, evaluating alignment and combinatorial effects of patterning, performing additional evaluation of the RNAseq data, and including more specific and complete data in supplementary data.

We appreciate the positive comments.